# FAST FINITE WIDTH NEURAL TANGENT KERNEL

## ABSTRACT

The Neural Tangent Kernel (NTK), defined as the outer product of the neural network (NN) Jacobians, $\Theta_\theta(x_1, x_2) = \left[\partial f(\theta, x_1)/\partial\theta\right]\left[\partial f(\theta, x_2)/\partial\theta\right]^T$, has emerged as a central object of study in deep learning. In the infinite width limit, the NTK can sometimes be computed analytically and is useful for understanding training and generalization of NN architectures. At finite widths, the NTK is also used to better initialize NNs, compare the conditioning across models, perform architecture search, and do meta-learning. Unfortunately, the finite width NTK is notoriously expensive to compute, which severely limits its practical utility.

We perform the first in-depth analysis of the compute and memory requirements for NTK computation in finite width networks. Leveraging the structure of neural networks, we further propose two novel algorithms that change the *exponent* of the compute and memory requirements of the finite width NTK, dramatically improving efficiency.

We open-source [github.com/iclr2022anon/fast_finite_width_ntk] our two algorithms as general-purpose JAX function transformations that apply to any differentiable computation (convolutions, attention, recurrence, etc.) and introduce no new hyper-parameters.

## 1 INTRODUCTION

The past few years have seen significant progress towards a theoretical foundation for deep learning. Much of this work has focused on understanding the properties of random functions in high dimensions. One significant line of work (Neal, 1994; Lee et al., 2018; Matthews et al., 2018; Novak et al., 2019; Garriga-Alonso et al., 2019; Hron et al., 2020; Yang, 2019) established that in the limit of infinite width, randomly initialized Neural Networks (NNs) are Gaussian Processes (called the NNGP). Building on this development, Jacot et al. (2018) showed that in function space the dynamics under gradient descent could be computed analytically using the so-called Neural Tangent Kernel (NTK) and Lee et al. (2019) showed that wide neural networks reduce to their linearization in weight space throughout training. A related set of results (Belkin et al., 2019; Spigler et al., 2019) showed that the ubiquitous bias-variance decomposition breaks down as high-dimensional models enter the so-called interpolating regime. Together these results describe learning in the infinite width limit and help explain the impressive generalization capabilities of NNs.

Insights from the wide network limit have had significant practical impact. The conditioning of the NTK has been shown to significantly impact trainability and generalization in NNs (Schoenholz et al., 2017; Xiao et al., 2018; 2020). This notion inspired initialization schemes like Fixup (Zhang et al., 2019), MetaInit (Dauphin & Schoenholz, 2019), and Normalizer Free networks (Brock et al., 2021a;b) and has enabled efficient neural architecture search (Park et al., 2020; Chen et al., 2021b). The NTK has additionally given insight into a wide range of phenomena such as: behavior of Generative Adversarial Networks (Franceschi et al., 2021), neural scaling laws (Bahri et al., 2021), and neural irradiance fields (Tancik et al., 2020). Kernel regression using the NTK has further enabled strong performance on small datasets (Arora et al., 2020), and applications such as dataset distillation (Nguyen et al., 2020; 2021) and uncertainty prediction (He et al., 2020; Adlam et al., 2020).

Despite the significant promise of theory based on the NTK, computing the NTK in practice is challenging. In the infinite width limit, the NTK can sometimes be computed analytically. However, it remains intractable for many architectures, and finite width corrections can be important to describe actual NNs used in practice. The NTK matrix can be computed for finite width networks as the outer

product of Jacobians using forward or reverse mode automatic differentiation (AD),

$$\underbrace{\Theta_\theta(x_1, x_2)}_{\mathbf{O} \times \mathbf{O}} := \underbrace{\left[\partial f(\theta, x_1)/\partial\theta\right]}_{\mathbf{O} \times \mathbf{P}} \underbrace{\left[\partial f(\theta, x_2)/\partial\theta\right]^T}_{\mathbf{P} \times \mathbf{O}}, \tag{1}$$

where $f$ is the forward pass NN function producing outputs in $\mathbb{R}^{\mathbf{O}}$, $\theta \in \mathbb{R}^{\mathbf{P}}$ are all trainable parameters, and $x_1$ and $x_2$ are two inputs to the network. If inputs are batches of sizes $\mathbf{N}_1$ and $\mathbf{N}_2$, the NTK is an $\mathbf{N}_1\mathbf{O} \times \mathbf{N}_2\mathbf{O}$ matrix.

Unfortunately, evaluating Eq. (1) is often infeasible due to time and memory requirements.

In this paper, we perform the first in-depth analysis of the compute and memory requirements for the NTK as in Eq. (1). Noting that forward and reverse mode AD are two extremes of a wide range of AD strategies (Naumann, 2004; 2008), we explore other methods for computing the NTK leveraging the structure of NNs used in practice. We propose two novel methods for computing the NTK that exploit different orderings of the computation. We describe the compute and memory requirements of our techniques in fully-connected (FCN) and convolutional (CNN) settings, and show that one is asymptotically more efficient in both settings. We compute the NTK over a wide range of NN architectures and demonstrate that these improvements are robust in practice. We open-source implementations of both methods as JAX function transformations.

## 2 RELATED WORK

The finite width NTK (denoted as simply NTK throughout this work) has been used extensively in many recent works, but to our knowledge implementation details and compute costs were rarely made public. Below we draw comparison to some of these works, but we stress that it only serves as a sanity check to make sure our contribution is valuable relative to the scale of problems that have been attempted (none of these works had efficient NTK computation as their central goal).

In order to compare performance of models based on the NTK and the infinite width NTK, Arora et al. (2019a, Table 2) compute the NTK of up to 20-layer, 128-channel CNN in a binary CIFAR-2 classification setting. In an equivalent setting with the same hardware (NVIDIA V100), we are able to compute the NTK of a 2048-channel CNN, i.e. a network with at least 256 times more parameters.

To demonstrate the stability of the NTK during training for wide networks, Lee et al. (2019, Figure S6) compute the NTK of up to 3-layer $2^{12}$-wide or 1-layer $2^{14}$-wide FCNs. In the same setting with the same hardware (NVIDIA V100), we can reach widths of at least $2^{14}$ and $2^{18}$ respectively, i.e. handle networks with at least 16 times more parameters.

To investigate convergence of a WideResNet WRN-28-$k$ (Zagoruyko & Komodakis, 2016) to its infinite width limit, Novak et al. (2020, Figure 2) evaluate the NTK of this model with widening factor $k$ up to 32. In matching setting and hardware, we are able to reach the widening factor of at least 64, i.e. work with models at least 4 times larger.

To meta-learn NN parameters for transfer learning in a MAML-like (Finn et al., 2017) setting, Zhou et al. (2021, Table 7) replace the inner training loop with NTK-based inference. They use up to 5-layer, 200-channel CNNs on MiniImageNet (Oreshkin et al., 2018) with scalar outputs and batch size 25. In same setting we achieve at least 512 channels, i.e. support models at least 6 times larger.

Park et al. (2020, §4.1) use the NTK to predict the generalization performance of architectures in the context of Neural Architecture Search (Zoph & Le, 2017, NAS); however, the authors comment on its high computational burden and ultimately use a different proxy. In another NAS setting, Chen et al. (2021a, §3.1.1) use the condition number of NTK to predict a model's trainability. Remarking its prohibitive cost, Chen et al. (2021b, Table 1) also use the NTK to evaluate the trainability of several ImageNet (Deng et al., 2009) models such as ResNet 50/152 (He et al., 2016), Vision Transformer (Dosovitskiy et al., 2021) and MLP-Mixer (Tolstikhin et al., 2021). However, in all of the above cases the authors only evaluate a pseudo-NTK, i.e. an NTK of a scalar-valued function,[1] which impacts the quality of the respective trainability/generalization proxy.

---

[1]Precisely, computing the Jacobian only for a single logit or the sum of all 1000 class logits. The result is not the full NTK, but rather a single diagonal block or the sum of its 1000 diagonal blocks (finite width NTK is a dense matrix, not block-diagonal).

| Method | Time | Memory | Use when |
|---|---|---|---|
| Jacobian contraction | $\mathbf{N}^2\mathbf{LO}^2\mathbf{W}^2$ | $\mathbf{NOW}^2 + \mathbf{N}^2\mathbf{O}^2 + \mathbf{NLW} + \mathbf{LW}^2$ | Don't |
| NTK-vector products | $\mathbf{N}^2\ \mathbf{O}^2\mathbf{W} + \mathbf{N}^2\mathbf{LOW}^2$ | $\mathbf{NOW}^2 + \mathbf{N}^2\mathbf{O}^2 + \mathbf{NLW} + \mathbf{LW}^2$ | $\mathbf{O} > \mathbf{W}$ or $\mathbf{N} = 1$ |
| Structured derivatives | $\mathbf{N}^2\mathbf{LO}^2\mathbf{W} + \mathbf{N}\ \mathbf{LOW}^2$ | $\mathbf{NOW} + \mathbf{N}^2\mathbf{O}^2 + \mathbf{NLW} + \mathbf{LW}^2$ | $\mathbf{O} < \mathbf{W}$ or $\mathbf{L} = 1$ |

Table 1: **Asymptotic time and memory cost of computing the NTK for an FCN.** Costs are for a pair of batches of inputs of size $\mathbf{N}$ each, and for $\mathbf{L}$-deep, $\mathbf{W}$-wide FCN with $\mathbf{O}$ outputs. Resulting NTK has shape $\mathbf{NO} \times \mathbf{NO}$. NTK-vector products allow a reduction of the time complexity, while Structured derivatives reduce both time and memory complexity. **Note:** presented are asymptotic cost estimates; in practice, all methods incur large constant multipliers (e.g. at least 3x for time; see §3.1). However, this generally does not impact the relative performance of different methods. See §3.6 for discussion, Table 7 for CNN, and Table 2 for more generic cost analysis.

| Method | Time | Memory | Use when |
|---|---|---|---|
| Jacobian contraction | $\mathbf{N}\ \mathbf{O}\,[\mathbf{FP}] + \mathbf{N}^2\mathbf{O}^2\mathbf{P}$ | $\mathbf{N}^2\mathbf{O}^2 + \mathbf{NO}\,[\mathbf{Y}^k + \mathbf{P}^l] + \mathbf{NY} + \mathbf{P}$ | $\mathbf{P} \ll \mathbf{Y}$, small $\mathbf{O}$, exotic primitives |
| NTK-vector products | $\mathbf{N}^2\mathbf{O}\,[\mathbf{FP}]$ | $\mathbf{N}^2\mathbf{O}^2 + \mathbf{NO}\,[\mathbf{Y}^k + \mathbf{P}^l] + \mathbf{NY} + \mathbf{P}$ | $\mathbf{FP} < \mathbf{OP}$, large $\mathbf{O}$, small $\mathbf{N}$ |
| Structured derivatives | $\mathbf{N}\ \mathbf{O}\,[\mathbf{FP}] + \mathbf{N}\ \mathbf{O}\ \mathbf{G} + \mathbf{N}\,[\mathbf{J} - \mathbf{OP}]$ | $\mathbf{N}^2\mathbf{O}^2 + \mathbf{NOY}^k + \mathbf{NJ}_l^k + \mathbf{NY} + \mathbf{P}$ | $\mathbf{FP} > \mathbf{OP}$, large $\mathbf{O}$, large $\mathbf{N}$ |

Table 2: **Asymptotic time and memory cost estimates of computing the NTK for a generic function.** $\mathbf{P}$ stands for the number of all parameters in the network, $\mathbf{Y}$ stands for size of all pre-activations in the network, $\mathbf{FP}$ stands for forward pass, and $\mathbf{G}$ and $\mathbf{J}$ depend on the structure of $\mathbf{FP}$ (§B). For example, FCNs usually have a cheap $\mathbf{FP} \leq \mathbf{OP}$, as it consists of a single matrix multiply with the parameter matrix, and therefore NTK-vector products are recommended. CNNs, notably when the number of output pixels $\mathbf{D}$ is large, have a costly $\mathbf{FP} \geq \mathbf{OP}$, since it amounts to $\mathbf{D}$ matrix multiplies with the parameters, and therefore Structured derivatives are preferred. For precise analysis, see Table 1 for FCN and Table 7 for CNN.

In contrast, in this work we can compute the full $1000 \times 1000$ NTK on the same models (1000 classes), i.e. perform a task 1000 times more costly.

Finally, we remark that in all of the above settings, scaling up by increasing width or by working with the true NTK (vs the pseudo-NTK) should lead to improved downstream task performance due to better infinite width/linearization approximation or higher-quality trainability/generalization proxy respectively, which makes our work especially relevant to modern research.

## 3 EFFICIENT FINITE WIDTH NTKS IN A SIMPLIFIED SETTING

To gain intuition for the problem, we start by analyzing and improving the cost of computing the NTK for a simple FCN. See §F for an equivalent analysis of CNNs. We summarize the resulting complexities for FCN in Table 1, CNN in Table 7, and a general takeaway in Table 2.

**Setting.** Consider an $\mathbf{L}$-layer FCN $f(\theta, x) = \theta^{\mathbf{L}} \phi\left(\theta^{\mathbf{L}-1} \dots \theta^1 \phi\left(\theta^0 x\right) \dots\right) \in \mathbb{R}^{\mathbf{O}}$, where $\mathbf{O}$ is the number of logits. We denote individual weight matrices as $\theta^l$ with shapes $\mathbf{W} \times \mathbf{W}$ (except for top-layer $\theta^{\mathbf{L}}$ of shape $\mathbf{O} \times \mathbf{W}$), where $\mathbf{W}$ is the width of the network, and write the set of all parameters as $\theta = \mathrm{vec}\left[\theta^0, \dots, \theta^{\mathbf{L}}\right] \in \mathbb{R}^{\mathbf{LW}^2 + \mathbf{OW}}$. We further define $x^l := \phi\left(y^{l-1}\right)$ as post-activations (with $x^0 := x$), and $y^l := \theta^l x^l$ as pre-activations with $y^{\mathbf{L}} = f(\theta, x)$. See Fig. 5 for a visual schematic of these quantities. For simplicity, we assume that inputs $x$ also have width $\mathbf{W}$, and $\mathbf{O} = \mathcal{O}(\mathbf{LW})$, i.e. the number of logits is dominated by the product of width and depth. In §L we repeat the same derivations without the latter assumption, and arrive at qualitatively identical conclusions.

The NTK of $f$ evaluated at two inputs $x_1$ and $x_2$ is an $\mathbf{O} \times \mathbf{O}$ matrix defined as

$$\Theta_\theta := \frac{\partial f(\theta, x_1)}{\partial \theta} \frac{\partial f(\theta, x_2)}{\partial \theta}^T = \sum_{l=0}^{\mathbf{L}} \frac{\partial f(\theta, x_1)}{\partial \theta^l} \frac{\partial f(\theta, x_2)}{\partial \theta^l}^T =: \sum_{l=0}^{\mathbf{L}} \Theta_\theta^l \in \mathbb{R}^{\mathbf{O} \times \mathbf{O}}, \qquad (2)$$

where we have defined $\Theta_\theta^l$ to be the summands. We omit dependence on $x_1$, $x_2$, and $f$ for brevity.

In §3.1 and §3.2 we describe the cost of several fundamental AD operations that we will use as building blocks throughout the text. We borrow the nomenclature introduced by Autograd (Maclau-

rin et al.) and describe Jacobian-vector products (JVP), vector-Jacobian products (VJP), as well as the cost of computing the Jacobian $\partial f(\theta, x)/\partial\theta$.

In §3.3, we describe the baseline complexity of evaluating the NTK, by computing two Jacobians and contracting them. This approach is used in most (likely all) prior works, and scales poorly with the NN width $\mathbf{W}$ and output size $\mathbf{O}$.

In §3.4 we present our first contribution, that consists in observing that many intermediate operations on weights performed by NNs possess a certain structure, that can allow linear algebra simplifications of the NTK expression, leading to a cheaper contraction and smaller memory footprint.

In §3.5 we present our second contribution, where we rephrase the NTK computation as instantiating itself row-by-row by applying the NTK-vector product function to columns of an identity matrix. As we will show, this trades off Jacobian contraction for more forward passes, which proves beneficial in many (but not all) settings.

## 3.1 JACOBIAN-VECTOR PRODUCTS AND VECTOR-JACOBIAN PRODUCTS

We begin by defining Jacobian-vector products and vector-Jacobian products:

$$\text{JVP}_{(f,\theta,x)} : \theta_t \in \mathbb{R}^{\mathbf{LW}^2+\mathbf{OW}} \mapsto \frac{\partial f(\theta, x)}{\partial\theta}\theta_t \in \mathbb{R}^{\mathbf{O}}, \tag{3}$$

$$\text{VJP}_{(f,\theta,x)} : f_c \in \mathbb{R}^{\mathbf{O}} \mapsto \frac{\partial f(\theta, x)}{\partial\theta}^T f_c \in \mathbb{R}^{\mathbf{LW}^2+\mathbf{OW}}. \tag{4}$$

The JVP can be understood as pushing forward a tangent vector in weight space to a tangent vector in the space of outputs; by contrast the VJP pulls back a cotangent vector in the space of outputs to a cotangent vector in weight space. These elementary operations correspond to forward and reverse mode AD respectively and serve as a basis for typical AD computations such as gradients, Jacobians, Hessians, etc.

Time and memory costs of JVP and VJP are asymptotically equivalent to the cost of the forward pass (**FP**), except for VJP additionally requires storing all intermediate activations. (see §N and Fig. 6).

For the case of FCNs, the time cost[2] of both operations is therefore

$$[\mathbf{FP}] = [\text{cost of all intermediate layers}] + [\text{cost of the top layer}] = \left[\mathbf{LW}^2\right] + \left[\mathbf{OW}\right] \sim \mathbf{LW}^2.$$

For a single input, the memory cost of computing both the JVP and the VJP are respectively,

$$[\text{size of all weights}] + [\text{size of activations at a single layer}] = \left[\mathbf{LW}^2 + \mathbf{OW}\right] + \left[\mathbf{W} + \mathbf{O}\right] \sim \mathbf{LW}^2,$$

$$[\text{size of all weights}] + [\text{size of activations in all layers}] = \left[\mathbf{LW}^2 + \mathbf{OW}\right] + \left[\mathbf{LW} + \mathbf{O}\right] \sim \mathbf{LW}^2.$$

Despite the fact that the VJP requires more memory to store intermediate activations, we see that for FCNs both computations are dominated by the cost of storing the weights.

**Batched inputs.** If $x$ is a batch of inputs of size $\mathbf{N}$, the time cost of JVP and VJP increases linearly to $\mathbf{NLW}^2$. The memory cost is slightly more nuanced. Since weights can be shared across inputs, the memory cost of the JVP and VJP are respectively,

$$[\text{size of all weights}] + \mathbf{N}[\text{size of activations at a single layer}]$$
$$= \left[\mathbf{LW}^2 + \mathbf{OW}\right] + \mathbf{N}[\mathbf{W} + \mathbf{O}] \sim \mathbf{LW}^2 + \mathbf{NW} + \mathbf{NO},$$
$$[\text{size of all weights}] + \mathbf{N}[\text{size of activations in all layers}] + \mathbf{N}[\text{size of all weight matrices}]$$
$$= \left[\mathbf{LW}^2 + \mathbf{OW}\right] + \mathbf{N}[\mathbf{LW} + \mathbf{O}] + \mathbf{N}\left[\mathbf{LW}^2 + \mathbf{OW}\right] \sim \mathbf{NLW}^2.$$

The cost of the VJP is dominated by the cost of storing the cotangents in weight space. However, for the purposes of computing the NTK, we will be contracting Jacobians layerwise and so we will only need to store one cotangent weight matrix, $\partial f/\partial\theta^l$, at a time. Thus, for the purposes of this work we end up with the following costs:

> - JVP costs $\mathbf{NLW}^2$ time and $\mathbf{LW}^2 + \mathbf{NW} + \mathbf{NO}$ memory.
> - VJP costs $\mathbf{NLW}^2$ time and $\mathbf{LW}^2 + \mathbf{NLW} + \mathbf{NW}^2 + \mathbf{NOW}$ memory.

---

[2]To declutter notation, we omit the $\mathcal{O}$ symbol to indicate asymptotic complexity in this work.

## 3.2 JACOBIAN COMPUTATION

For neural networks, the Jacobian is most often computed by evaluating the VJP on rows of the identity matrix $I_\mathbf{O}$, i.e.

$$\left[\partial f\left(\theta, x\right)/\partial\theta\right]^T = \left[\partial f\left(\theta, x\right)/\partial\theta\right]^T I_\mathbf{O} \in \mathbb{R}^{\left(\mathbf{LW}^2 + \mathbf{OW}\right)\times\mathbf{O}}. \tag{5}$$

It follows that computing the Jacobian takes $\mathbf{O}$ evaluations of the VJP. However, as mentioned in §3.1, we only need to store one $\partial f/\partial\theta^l$ at a time, while the weights and intermediate activations are reused across evaluations. Thus, time and memory costs to compute the Jacobian are respectively,

$$\mathbf{ON}\left(\left[\text{cost of all intermediate layers}\right] + \left[\text{cost of the top layer}\right]\right)$$
$$= \mathbf{ON}\left(\left[\mathbf{LW}^2\right] + \left[\mathbf{OW}\right]\right) \sim \mathbf{NLOW}^2 + \mathbf{NO}^2\mathbf{W},$$

$$\left[\text{size of all weights}\right] + \mathbf{N}\left[\text{size of activations in all layers}\right] + \mathbf{ON}\left[\text{size of a single weight matrix}\right]$$
$$= \left[\mathbf{LW}^2 + \mathbf{OW}\right] + \mathbf{N}\left[\mathbf{LW} + \mathbf{O}\right] + \mathbf{ON}\left[\mathbf{W}^2 + \mathbf{OW}\right] \sim \mathbf{LW}^2 + \mathbf{NLW} + \mathbf{NOW}^2 + \mathbf{NO}^2\mathbf{W}.$$

Therefore, asymptotically,

> Jacobian costs $\mathbf{NLOW}^2 + \mathbf{NO}^2\mathbf{W}$ time and $\mathbf{LW}^2 + \mathbf{NLW} + \mathbf{NOW}^2 + \mathbf{NO}^2\mathbf{W}$ memory.

## 3.3 JACOBIAN CONTRACTION

We now analyze the cost of computing the NTK, starting with the direct computation as the product of two Jacobians. Consider a single summand from Eq. (2):

$$\underbrace{\Theta_\theta^l}_{\mathbf{O}\times\mathbf{O}} = \underbrace{\frac{\partial f\left(\theta, x_1\right)}{\partial\theta^l}}_{\mathbf{O}\times\left(\mathbf{W}\times\mathbf{W}\right)} \underbrace{\frac{\partial f\left(\theta, x_2\right)}{\partial\theta^l}^T}_{\left(\mathbf{W}\times\mathbf{W}\right)\times\mathbf{O}}. \tag{6}$$

The time cost of this contraction is $\mathbf{O}^2\mathbf{W}^2$, and the memory necessary to instantiate each factor and the result is $\mathbf{OW}^2 + \mathbf{O}^2$. Repeating the above operation for each $\theta^l$, we arrive at $\mathbf{LO}^2\mathbf{W}^2$ time cost and unchanged memory, due to being able to process summands sequentially.

**Batched inputs.** If we consider $x_1$ and $x_2$ to be input batches of size $\mathbf{N}$, then the resulting NTK is a matrix of shape $\mathbf{NO}\times\mathbf{NO}$, and the time cost becomes $\mathbf{N}^2\mathbf{LO}^2\mathbf{W}^2$, while memory grows to $\left[\text{NTK matrix size}\right] + \left[\text{factors size}\right] = \mathbf{N}^2\mathbf{O}^2 + \mathbf{NOW}^2$.

What remains is to account for the cost of computing and storing individual cotangents $\partial f/\partial\theta^l$, which is exactly the cost of computing the Jacobian (§3.2). Adding the costs up we obtain

> Jacobian contraction costs $\mathbf{N}^2\mathbf{LO}^2\mathbf{W}^2$ time and $\mathbf{N}^2\mathbf{O}^2 + \mathbf{NOW}^2 + \mathbf{NO}^2\mathbf{W} + \mathbf{LW}^2 + \mathbf{NLW}$ memory.

## 3.4 LEVERAGING STRUCTURED DERIVATIVES FOR COMPUTING THE NTK

We can rewrite $\Theta^l_\theta$ in Eq. (6) using the chain rule and our pre- and post-activation notation as:

$$\Theta_\theta^l = \left[\frac{\partial f\left(\theta, x_1\right)}{\partial y_{x_1}^l}\frac{\partial y_{x_1}^l}{\partial\theta^l}\right]\left[\frac{\partial f\left(\theta, x_2\right)}{\partial y_{x_2}^l}\frac{\partial y_{x_2}^l}{\partial\theta^l}\right]^T = \underbrace{\frac{\partial f\left(\theta, x_1\right)}{\partial y_{x_1}^l}}_{\mathbf{O}\times\mathbf{W}}\underbrace{\frac{\partial y_{x_1}^l}{\partial\theta^l}}_{\mathbf{W}\times\left(\mathbf{W}\times\mathbf{W}\right)}\underbrace{\frac{\partial y_{x_2}^l}{\partial\theta^l}^T}_{\left(\mathbf{W}\times\mathbf{W}\right)\times\mathbf{W}}\underbrace{\frac{\partial f\left(\theta, x_2\right)}{\partial y_{x_2}^l}^T}_{\mathbf{W}\times\mathbf{O}}. \tag{7}$$

At face value, rewriting Eq. (6) this way is unhelpful as it appears to have introduced additional costly contractions. However, recall that $y^l = \theta^l x^l$, and therefore

$$\frac{\partial y_{x_1}^l}{\partial\theta^l} = I_\mathbf{W}\otimes x_1^l{}^T, \quad \frac{\partial y_{x_2}^l}{\partial\theta^l} = I_\mathbf{W}\otimes x_2^l{}^T, \tag{8}$$

where $\otimes$ is the Kronecker product. Plugging Eq. (8) into Eq. (7) we obtain (see §G)

$$\Theta_\theta^l = \left( \underbrace{x_1^l{}^T}_{1 \times \mathbf{W}} \quad \underbrace{x_2^l}_{\mathbf{W} \times 1} \right) \left[ \underbrace{\frac{\partial f(\theta, x_1)}{\partial y_{x_1}^l}}_{\mathbf{O} \times \mathbf{W}} \quad \underbrace{\frac{\partial f(\theta, x_2)}{\partial y_{x_2}^l}{}^T}_{\mathbf{W} \times \mathbf{O}} \right], \tag{9}$$

and observe that it takes only $\mathbf{O}^2\mathbf{W}$ time and $\mathbf{OW} + \mathbf{O}^2$ memory. Accounting for depth, time cost becomes $\mathbf{LO}^2\mathbf{W}$, while memory does not change since the summands can be processed sequentially.

**Batched inputs.** The time cost grows quadratically with the bath size $\mathbf{N}$ up to $\mathbf{N}^2\mathbf{LO}^2\mathbf{W}$, while the memory cost increases to $\mathbf{N}^2\mathbf{O}^2 + \mathbf{NOW}$ to store the resulting NTK and $\partial f(\theta, x) / \partial y_x^l$ factors.

Finally, we need to account for the cost of computing the derivatives, $\partial f / \partial y^l$, and post-activations, $x^l$. Notice that both $x^l$ and $\partial f / \partial y^l$ arises naturally when computing the Jacobian as the primals and cotangents in layer $l$ respectively. However, since we do not need to compute the weight space cotangents explicitly (i.e. we cut the backpropagation algorithm short) the memory cost will be,

$$[\text{size of all weights}] + \mathbf{N} [\text{size of activations in all layers}]$$
$$= \left[ \mathbf{LW}^2 + \mathbf{OW} \right] + \mathbf{N} \left[ \mathbf{LW} + \mathbf{O} \right] \sim \mathbf{LW}^2 + \mathbf{NLW}.$$

The extra time cost is asymptotically the cost of $\mathbf{O}$ forward passes, $\mathbf{NLOW}^2$ which is the same as the Jacobian. However, as we will see in experiments, in practice we'll often compute the NTK faster than the Jacobian due to not computing the weight space cotangents $\partial f / \partial \theta^l$. Altogether,

> By leveraging Structured derivatives in NN computations, we have reduced the cost of NTK to $\mathbf{N}^2\mathbf{LO}^2\mathbf{W} + \mathbf{NLOW}^2$ time and $\mathbf{N}^2\mathbf{O}^2 + \mathbf{NOW} + \mathbf{LW}^2 + \mathbf{NLW}$ memory.

The key insight was to leverage the constant block-diagonal structure of the pre-activation derivatives $\partial y^l / \partial \theta^l$. This idea is quite general; as we discuss in §4 and detail in the appendix, similar structure exists for many common operations such as convolutions, pooling, and arithmetic.

We highlight that these computational improvements do not emerge automatically in AD. While JAX and other libraries leverage structures analogous to Eq. (8) to efficiently compute single evaluations of the VJP and JVP, this knowledge is lost once the (structureless) Jacobian $\partial f(\theta, x_1) / \partial \theta^l$ is instantiated, and cannot be taken advantage of in the following contraction with $\partial f(\theta, x_2) / \partial \theta^l$. We discuss how we impose this structure to compute the NTK for general neural networks in §4.

### 3.5 NTK VIA NTK-VECTOR PRODUCTS

Computing the Jacobian contraction using Jacobian first instantiates the Jacobian using using VJPs and then performs a contraction. Structured derivatives use a similar strategy, but speed-up the contraction and avoid explicitly instantiating the weight space cotangents. Here we avoid performing a contraction altogether at the cost of extra VJP/JVP calls; this ends up being beneficial for FCNs.

We introduce the linear function performing the NTK-vector product: $\Theta\text{VP} : v \in \mathbb{R}^{\mathbf{O}} \mapsto \Theta_\theta v \in \mathbb{R}^{\mathbf{O}}$. Applying this function to $\mathbf{O}$ columns of the identity matrix $I_{\mathbf{O}}$ allows us to compute the NTK, i.e. $\Theta_\theta I_{\mathbf{O}} = \Theta_\theta$. The cost of evaluating the NTK in this fashion is equal to $\mathbf{O}$ times the cost of a single NTK-vector product evaluation $\Theta\text{VP}(v)$. We now expand $\Theta\text{VP}(v) = \Theta_\theta v$ as

$$\frac{\partial f(\theta, x_1)}{\partial \theta} \frac{\partial f(\theta, x_2)}{\partial \theta}^T v = \frac{\partial f(\theta, x_1)}{\partial \theta} \text{VJP}_{(f, \theta, x_2)}(v) = \text{JVP}_{(f, \theta, x_1)} \left[ \text{VJP}_{(f, \theta, x_2)}(v) \right], \tag{10}$$

where we have observed that, if contracted from right to left, the NTK-vector product can be expressed as a composition of a JVP and VJP of the underlying function $f$. The cost of this operation is asymptotically equivalent to the cost of the Jacobian, since it consists of $\mathbf{O}$ VJPs followed by $\mathbf{O}$ (cheaper) JVPs. Therefore it costs $\mathbf{LOW}^2 + \mathbf{O}^2\mathbf{W}$ time and $\mathbf{LW}^2 + \mathbf{OW}^2 + \mathbf{O}^2\mathbf{W}$ memory.

**Batched inputs.** In the batched setting Eq. (10) is repeated for each pair of inputs, and therefore time increases by a factor of $\mathbf{N}^2$ to become $\mathbf{N}^2\mathbf{LOW}^2 + \mathbf{N}^2\mathbf{O}^2\mathbf{W}$. However, the memory cost grows only linearly in $\mathbf{N}$ (except for the cost of storing the NTK of size $\mathbf{N}^2\mathbf{O}^2$), since intermediate activations and derivatives necessary to compute the JVP and VJP can be computed for each batch $x_1$ and $x_2$

separately; these quantities are then reused for every pairwise combination resulting in a memory equivalent to the Jacobian, i.e. $\mathbf{N^2O^2} + \left(\mathbf{LW^2} + \mathbf{NOW^2} + \mathbf{NO^2W} + \mathbf{NLW}\right)$, resulting in

> NTK computation as a sequence of NTK-vector products costs $\mathbf{N^2LOW^2} + \mathbf{N^2O^2W}$ time and $\mathbf{N^2O^2} + \mathbf{NOW^2} + \mathbf{LW^2} + \mathbf{NLW}$ memory.

### 3.6 SUMMARY

Structured derivatives and NTK-vector products allow a reduction in the time cost of NTK computation in different ways, and Structured derivatives also reduce memory requirements. Structured derivatives are beneficial for wide networks, with large $\mathbf{W}$, and NTK-vector products are beneficial for networks with large outputs $\mathbf{O}$. We confirm our predictions with FLOPs measurements in Fig. 1.

We further confirm our methods can provide orders of magnitude speed-ups and memory savings on all major hardware platforms in Fig. 1 (right) and Fig. 3. However, we notice that our wall-clock time measurements often deviate from predictions due to unaccounted constant overheads of various methods, hardware specifics, padding, and the (largely black-box) behavior of the XLA compiler. Notably, in practice, we find Structured derivatives almost always outperform NTK-vector products.

Finally, we evaluate our methods in the wild, and confirm computational benefits on full ImageNet models in Fig. 2 (ResNets, He et al. (2016)) and Fig. 4 (WideResNets, Zagoruyko & Komodakis (2016); Vision Transformers and Transformer-ResNet hybrids Dosovitskiy et al. (2021); Steiner et al. (2021); and MLP-Mixers Tolstikhin et al. (2021)). Computing the full $\mathbf{O} \times \mathbf{O} = 1000 \times 1000$ NTK for many of these models on modern accelerators is only possible with Structured derivatives.

## 4 STRUCTURED DERIVATIVES FOR GENERIC FUNCTIONS

Here we generalize the idea of leveraging structure in subexpressions presented in §3.4. This section (and our implementation) is not specific to NNs and applies to any differentiable function.

Consider two differentiable functions defined on a common input domain:

$$f_i : \left(\theta^0, \ldots, \theta^{\mathbf{L}}\right) \in \mathbb{R}^{\mathbf{P}_0 \times \cdots \times \mathbf{P}_{\mathbf{L}}} \mapsto f_i\left(\theta^0, \ldots, \theta^{\mathbf{L}}\right) \in \mathbb{R}^{\mathbf{O}_i} \quad (i \in \{1, 2\}).$$

For NNs, typically $\left(\theta^0, \ldots, \theta^{\mathbf{L}}\right)$ correspond to trainable parameters in layers $0, \ldots, \mathbf{L}$, and $f_i\left(\theta^0, \ldots, \theta^{\mathbf{L}}\right) = f\left(\theta^0, \ldots, \theta^{\mathbf{L}}, x_i\right)$, $x_i$ being network inputs, $\mathbf{O}_i = \mathbf{O}$ being the number of outputs (logits, classes). The NTK is defined as

$$\Theta_\theta\left(f_1, f_2\right) := \sum_{l=0}^{\mathbf{L}} \frac{\partial f_1}{\partial \theta^l} \frac{\partial f_2}{\partial \theta^l}^T \in \mathbb{R}^{\mathbf{O}_1 \times \mathbf{O}_2}. \tag{11}$$

Assume the following decompositions of $f_i$ into computational graphs made of primitives $y_i$:

$$f_i\left(\theta^0, \ldots, \theta^{\mathbf{L}}\right) = \tilde{f}_i\left(y_i^1(\theta^0, \ldots, \theta^{\mathbf{L}}), \ldots, y_i^{\mathbf{K}_i}(\theta^0, \ldots, \theta^{\mathbf{L}})\right) \quad (i \in \{1, 2\}). \tag{12}$$

with $y_i^k\left(\theta^0, \ldots, \theta^{\mathbf{L}}\right) \in \mathbb{R}^{\mathbf{Y}_i^k}$. In common NNs, $y_i^{k_i}$ would correspond to pre-activations evaluated on inputs $x_i$ in layer $k_i$, and, without weight sharing, typically $\mathbf{K}_1 = \mathbf{K}_2 = \mathbf{L}$. However, we do not impose any relationship between the number of parameter variables $\mathbf{L}$ and number of primitives $\mathbf{K}_1$ and $\mathbf{K}_2$, allowing arbitrary weight sharing. We can then use the chain rule in Eq. (2) to obtain:

$$\Theta_\theta\left(f_1, f_2\right) = \sum_{l, k_1, k_2}^{\mathbf{L}, \mathbf{K}_1, \mathbf{K}_2} \left(\frac{\partial \tilde{f}_1}{\partial y_1^{k_1}} \frac{\partial y_1^{k_1}}{\partial \theta^l}\right) \left(\frac{\partial \tilde{f}_2}{\partial y_2^{k_2}} \frac{\partial y_2^{k_2}}{\partial \theta^l}\right)^T = \sum_{l, k_1, k_2}^{\mathbf{L}, \mathbf{K}_1, \mathbf{K}_2} \frac{\partial \tilde{f}_1}{\partial y_1^{k_1}} \frac{\partial y_1^{k_1}}{\partial \theta^l} \frac{\partial y_2^{k_2}}{\partial \theta^l}^T \frac{\partial \tilde{f}_2}{\partial y_2^{k_2}}^T. \tag{13}$$

All methods from §3 perform the sum of contractions in Eq. (13) one way or another. Jacobian contraction uses VJPs to implicitly contract each summand "outside-in", i.e. it first computes $\partial f_i/\partial \theta^l$ terms with VJPs followed by their contraction. As discussed in §3.3, this costs $\mathbf{NO}\left[\mathbf{FP}\right] + \mathbf{N^2O^2P}$.

NTK-vector products use both JVPs and VJPs to contract "Right-to-left", i.e. first compute $\partial f_2/\partial \theta^l$ as an implicit contraction of $\partial f_2/\partial y_2$ with $\partial y_2/\partial \theta^l$ via VJP, followed by an implicit contraction of the result with $\partial y_1/\partial \theta^l$ via a JVP, followed by another implicit contraction with $\partial f_1/\partial y_1$ with another JVP. Per §3.5 this costs $\mathbf{N^2O}\left[\mathbf{FP}\right]$.

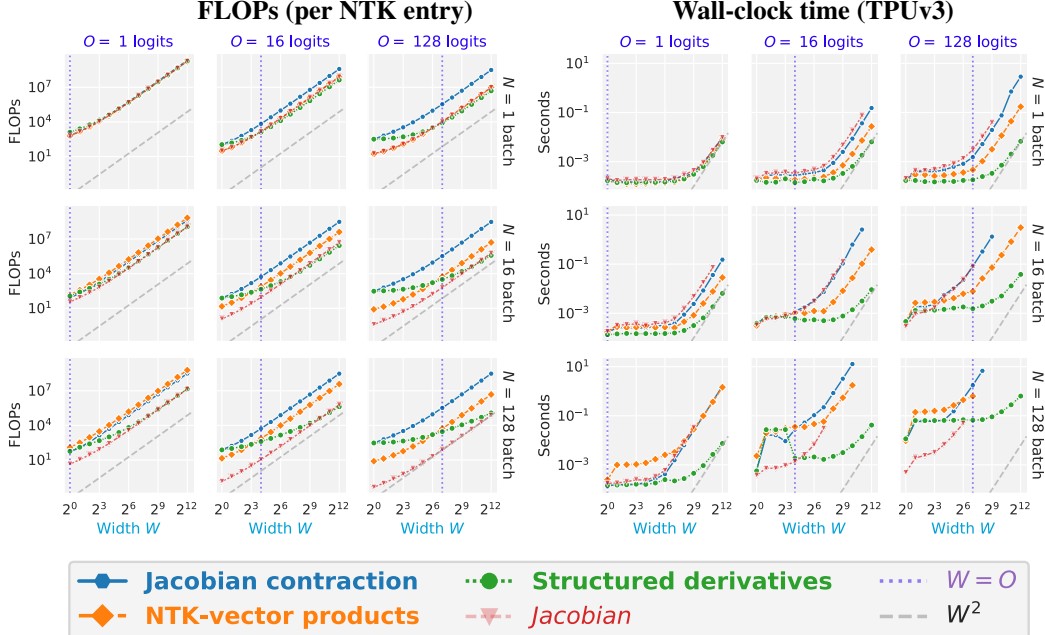

Figure 1: **FLOPs (left) and wall-clock time (right) of computing the NTK for a 10-layer ReLU FCN.** As predicted by Table 1, our methods almost always outperform Jacobian contraction, allowing orders of magnitude speed-ups and memory improvements for realistic problem sizes. **FLOPs per NTK entry:** We confirm several specific predictions: (1) NTK-vector products are the best performing method for $N = 1$, and have cost equivalent to Jacobian for any width $W$ or output size $O$ (top row); (2) NTK-vector products offer an $O$-fold improvement over Jacobian contraction (left to right columns); (3) NTK-vector products are equivalent to Jacobian contraction for $O = 1$ (leftmost column); (4) Structured derivatives outperform NTK-vector products iff $O < W$ ($O = W$ are plotted as pale vertical lines, which is where Structured derivatives and NTK-vector products intersect); (5) Structured derivatives approach the cost of Jacobian in the limit of large width $W$ (left to right). (6) All methods, as expected, scale quadratically with width $W$. **Wall-clock runtime:** In real applications, given hardware-specific constraints, padding, and delicate interplay with the XLA compiler, we observe that: (1) NTK-vector products improve upon Jacobian contraction for $O > 1$, but the effect is not perfectly robust (see bottom row for small $W$ and Fig. 3, notably GPU platforms); (2) Structured derivatives robustly outperform all other methods, including simply computing the Jacobian, as discussed in §3.4; (3) Structured derivatives have lower memory footprint, and reach up to 8x larger widths (bottom right; missing points indicate out-of-memory), i.e. can process models up to 64x larger than other methods, as discussed in §3.4; (4) All methods have a smaller memory footprint than Jacobian (see §3.1). **More:** Fig. 3 for other hardware platforms, §H for details.

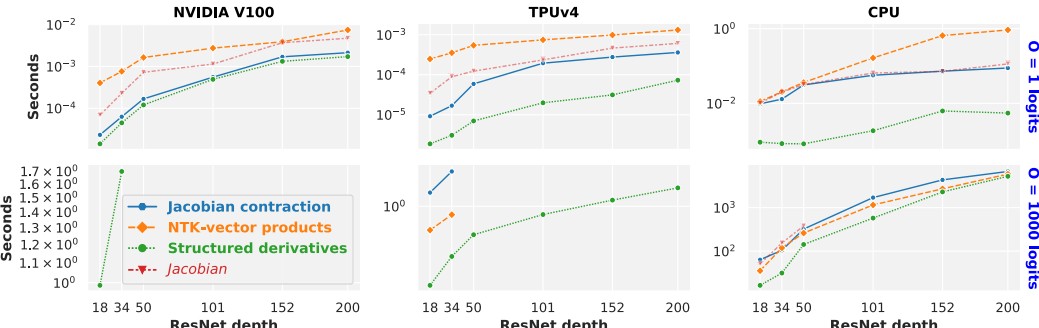

Figure 2: **Wall-clock time cost of computing an NTK for several ResNet sizes on a pair of ImageNet inputs.** Structured derivatives allow the NTK to be computed faster and for larger models (see bottom row – missing points indicate out-of-memory). NTK-vector products, as predicted in §3.6 and Table 2, are advantageous for large $O$ (bottom row), but are suboptimal when the cost of the forward pass is large relative to the number of parameters, e.g. when there is a lot of weight sharing (see Table 7 and Table 2), which is the case for convolutions. See Fig. 4 for more ImageNet models, §F for analysis of CNN NTK computational complexity, and §H for experimental details.

However, recall from §3.4, while JVPs and VJPs themselves are computationally optimal, higher-order computations like their contraction (Jacobian contraction) or composition (NTK-vector products) are generally not. The idea of Structured derivatives is to design rules for efficient computation of such contractions, similarly to how JAX and has rules for efficient JVPs and VJPs. From Eq. (13), in the general case this requires hand-made rules for all pairwise combinations of primitives $y_1$ and $y_2$. Due to quadratic scaling in the number of primitives, we restrict the current implementation to rules that operate on individual primitives $y$. This still provides substantial computational benefit.

Specifically, our rules identify a few simple types of structure (e.g. block diagonal, constant-block diagonal, tiling) in $\partial y / \partial \theta^l$, that allow us to simplify the contraction in Eq. (13). In practice this amounts to replacing the inner terms $\partial y_1^{k_1} / \partial \theta^l$ and $\partial y_2^{k_2} / \partial \theta^l$ with their (much) smaller subarrays, and modifying the instructions passed to `np.einsum` that contracts all 4 terms. In §C we provide specific descriptions of our rules and their impact on the computational complexity of Eq. (13).

In Table 1 and Table 7 we show that our rules are asymptotically better than Jacobian contraction for matrix multiplications and convolutions, and verify that they are practically beneficial in a much wider set of operations used by contemporary ImageNet models in Fig. 2 and Fig. 4.

For both Structured derivatives and NTK-vector products a fully general and rigorous comparison of complexities is not feasible since it will rely upon specifics of the model architecture and the pairs of primitives, $y_1$ and $y_2$, present in the network. Nonetheless, we can offer heuristics that suggest when each method will be beneficial. The time complexity of Structured derivatives has the form of $\mathbf{NO}\,[\mathbf{FP}] + \mathbf{NOG} + \mathbf{N}\,[\mathbf{J} - \mathbf{OP}]$, where $\mathbf{G}$ is related to the cost of contraction, and $\mathbf{J}$ to the cost of computing $\partial y / \partial \theta^l$ (exact values depend on the structure present in $y_1$ and $y_2$). This is guaranteed to be no worse than Jacobian contraction for FCNs and CNNs. From Table 2, the performance of NTK-vector products relative to Jacobian contraction ultimately depends on the cost of the forward pass through the network, $[\mathbf{FP}]$, relative to $\mathbf{OP}$. In practice this amounts to best performance on models without weight sharing like FCNs.

Owing to the nuanced trade-offs between different computational methods in the general case, we release all our implementations as a single function that allows the user to manually select the desired implementation. For convenience, we include an automated setting which will perform FLOPs analysis for each method at compilation time and automatically select the most efficient one.

## 5 IMPLEMENTATION

Both algorithms are implemented in JAX (Bradbury et al., 2018) as the following function transformation `ntk_fn` : $[f : (\theta, x) \mapsto f(\theta, x)] \mapsto [\Theta : (x_1, x_2, \theta) \mapsto \Theta_\theta(x_1, x_2)]$, i.e. our function accepts any function $f$ with the above signature and returns the efficient NTK kernel function operating on inputs $x_1$ and $x_2$ and parameterized by $\theta$. Inputs $x$, parameters $\theta$, and outputs $f(\theta, x)$ can be arbitrary PyTrees. We rely on many utilities from JAX and Neural Tangents (Novak et al., 2020).

**NTK-vector products** algorithm is implemented by using JAX core operations such as `vjp`, `jvp`, and `vmap` to map the NTK-vp function to the $I_\mathbf{O}$ matrix and to parallelize the computation over pairwise combinations of $\mathbf{N}$ inputs in each batch $x_1$ and $x_2$.

**Structured derivatives** algorithm is implemented as a Jaxpr interpreter, built on top of the default JAX reverse mode AD interpreter. On a high level, the algorithm performs the sum in Eq. (13). Each summand is a contraction of 4 factors: $\partial \tilde{f}_1 / \partial y_1, \partial y_1 / \partial \theta, \partial y_2 / \partial \theta, \partial \tilde{f}_2 / \partial y_2$.

First, we linearize $f$ to obtain a computational graph constructed out of a limited set (54,[3] see Table 5) of linear primitives $y^1, \ldots, y^\mathbf{K}$. Then, we can obtain two factors $\partial \tilde{f}_1 / \partial y_1, \partial \tilde{f}_2 / \partial y_2$ as part of a backward pass almost identical to calling `jax.jacobian` $(f)(\theta, x)$. To contract these terms with $\partial y_1 / \partial \theta$ and $\partial y_2 / \partial \theta$, as described above, we query a dictionary of rules which map primitives to a structural description (§C.8); for a given pair of primitives, these rules allow us to analytically simplify the contraction and avoid explicitly instantiating the derivatives.

---

[3]JAX leverages a similar approach to implement only 54 transpose rules for linear primitives for reverse mode differentiation instead of 131 VJP rules (Frostig et al., 2021).

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
