# OpenReview forum: "Fast Finite Width Neural Tangent Kernel"
_ICLR.cc/2022/Conference — ICLR 2022 Submitted_

### Official Review · Reviewer_Ye97 · 2021-10-28

**Correctness:** 3
**Technical Novelty And Significance:** 3
**Empirical Novelty And Significance:** 4
**Recommendation:** 6
**Confidence:** 4

**Main Review:**

Strengths

(1) This work provides a comprehensive study on various ways to compute the finite-width NTK. In particular, the time cost and memory cost are given for each case, providing a detailed reference for researchers in related fields.

(2) The author makes the article easy to understand through careful structural planning.

Weakness:

(1) I am confused with the setting that ${\rm O} = O( {\rm L W})$, for which the authors demonstrated that the number of logits is dominated by the product of width and depth.  If I understand correctly, $\rm O$ is the number of neurons in the last layer of the neural network. Thus it should be independent with ${\rm W}$ and ${\rm L}$.

(2) The dashed lines in Figure 1 are hard to distingish, I suggest the authros could make it clearer.

**Summary Of The Paper:**

This work aims to solve the computation problem of the finite-width neural tangent kernel (NTK), which is a central object in deep learning. The authors analyze the computation and memory requirements for finite-width NTK and propose two novel algorithms that can improve efficiency. Open-source has been provided by the authors.

**Summary Of The Review:**

Overall, this article is clearly written and well structured. If the author can solve the problem of unclear dotted line in Figure 1, it will be more perfect.

Besides, this work builds on JAX. However, the most popular deep learning framework used is PyTorch and Tensorflow. It would be better if the authors discuss why they choose JAX instead of other frameworks.

---

> ### Author Response · Authors · 2021-11-13
> **[Part 2 / 2] JAX vs Tensorflow/PyTorch**
>
> > Besides, this work builds on JAX. However, the most popular deep learning framework used is PyTorch and Tensorflow. It would be better if the authors discuss why they choose JAX instead of other frameworks.
>
> Thank you for the suggestion, we will elaborate on this in the next revision. There were several reasons why we chose JAX for our initial implementation:
>
> 1. Structured derivatives rely on manual implementation of structure rules for primitives present in the computation graph of a function. JAX has a small primitive set of about 131 primitives, while PyTorch has more than 400, and Tensorflow appears to have even more (see [1, section 2], also https://www.tensorflow.org/api_docs/python/tf/raw_ops vs https://github.com/pytorch/pytorch/blob/master/tools/autograd/derivatives.yaml vs  https://jax.readthedocs.io/en/latest/jax.lax.html). Further, by leveraging `jax.linearize`, we reduce our task to implementing structure rules for _only linear_ primitives, of which JAX has only 54 (This follows from the fact that the NTK of a function is equal to the NTK of the linearized function at the same primal parameters $\theta$; see also [1, section 1], for how JAX uses the same insight to not implement all 131 VJP rules, but only implement 54 transpose rules for reverse-mode AD). To our knowledge neither PyTorch nor Tensorflow have an equivalent transformation, and hence JAX was a natural choice due to the very concise set of primitives that we need to handle.
>
> 1. NTK-vector products critically rely on forward-mode automatic differentiation (JVP), and Structured derivatives also use it (albeit it’s not crucial). PyTorch appears to not have an efficient forward-mode AD (see https://pytorch.org/docs/stable/generated/torch.autograd.functional.jvp.html#torch.autograd.functional.jvp – _“The jvp is currently computed by using the backward of the backward (sometimes called the double backwards trick) as we don’t have support for forward mode AD in PyTorch at the moment.”_). This means that, at least for the moment, NTK-vector products cannot be efficiently implemented in PyTorch.
>
> 1. Structured derivatives rely crucially on our ability to traverse the computation graph in the form of a Jaxpr to rewrite contractions using our substitution rules. While this is possible in PyTorch using the new `pytorch.fx` tracing machinery, it is more difficult to do so in TensorFlow. In TF one must traverse and rewrite the computation graph which is typically messier than writing a Jaxpr interpreter.
>
> 1. All implementations (even Jacobian contraction) rely heavily on `jax.vmap` (and we believe, in many cases, it is indispensable). While PyTorch has [released](https://github.com/pytorch/pytorch/issues/42368) a prototype of `vmap` in May 2021, it was not available when we started this project (more than a year ago, with initial prototypes dating back all the way to 2019).
>
> For the reasons above, implementing our algorithms in other frameworks would be quite challenging. However, if there is strong demand for it, and the missing features make it into TF/PyTorch, we would be happy to reconsider bringing our algorithms to other libraries. In the near term, one option is to use something like https://github.com/dmlc/dlpack to do zero-cost data transfer between JAX and PyTorch, allowing users to mix the two frameworks. Additionally, the [`jax2tf`](https://github.com/google/jax/tree/main/jax/experimental/jax2tf) transformation already allows researchers to round-trip computations between TensorFlow and JAX, so it should be possible to use our library to perform fast evaluation of the NTK on networks defined using TensorFlow.
>
>
> ### References
>
> [1] [Decomposing reverse-mode automatic differentiation](https://arxiv.org/abs/2105.09469), Frostig et al., 2021.

---

> > ### Comment · Reviewer_Ye97 · 2021-11-30
> > **Thanks for your response**
> >
> > I would like to thank the authors for their detailed response. The clarification on complexity analysis and comparison between Jax and Tensorflow and PyTorch regarding the computation of NTK has addressed my concerns. Overall, I think this work is marginally above the acceptance threshold.

---

> ### Author Response · Authors · 2021-11-13
> **[Part 1 / 2] Minor clarifications**
>
> Thank you for your thoughtful review and suggestions! Please find our replies below.
>
>
> > (1) I am confused with the setting that O=O(LW), for which the authors demonstrated that the number of logits is dominated by the product of width and depth. If I understand correctly, O is the number of neurons in the last layer of the neural network. Thus it should be independent with W and L.
>
> You are correct, $O$ is the number of top-layer neurons (#classes, #logits; for example, 10 for CIFAR-10, 1000 for ImageNet). It is generally independent of width $W$ and depth $L$, and our complexities are computed as a function of  $O$ along with $L$ and $W$. However, in the vast majority of neural network architectures $O < L W$, and we have only made the assumption that $O = \mathcal{O}\left(L W\right)$ (i.e. $O$ is asymptotically bounded by $L W$) to make the derivations and complexity estimates more concise. Precisely, without this assumption Jacobian contraction and Structured derivatives would gain extra $N^2 O^3 W$ and $N^2 O^3$ time complexity terms respectively, but this would not affect any of our conclusions, and would be inconsequential in practice since usually $O < L W$, and this term would be dominated by other terms. We will add this full complexity analysis to the Appendix in the next revision.
>
>
> > (2) The dashed lines in Figure 1 are hard to distingish, I suggest the authros could make it clearer.
>
> > If the author can solve the problem of unclear dotted line in Figure 1, it will be more perfect.
>
> Good point, we will fix it in the next revision.

---

### Official Review · Reviewer_uyYg · 2021-10-31

**Correctness:** 4
**Technical Novelty And Significance:** 1
**Empirical Novelty And Significance:** 2
**Recommendation:** 3
**Confidence:** 5

**Main Review:**

The paper describes three methods for computing the neural tangent kernel for a given a batch of vectors and a feed-forward neural network. The first method, Jacobian contraction is quite straightforward and is not suggested for use. The second one,  NTK-vector products is based on reducing the task to the computation of Jacobian-vector and vector-Jacobian products (which themselves can be computed in a fashion similar to forward pass). The method is suggested for use for networks whose width is smaller than the dimension of the output space (probably for contractive autoencoders). The third method, Structured derivatives, is based on a simple identity (9) that simplifies the computation of NTK over weights of the same layer. It turns out that this method is preferable for shallow networks or the typical sitution when  width is larger than the dimension of output space.

Experiments are convincing. The provided code is helpful for researchers who need a fast computation of NTK. Though the ideological (mathematical part) is very simple.

**Summary Of The Paper:**

Experiments are convincing. The provided code is helpful for researchers who need a fast computation of NTK. Though the ideological (mathematical part) is very simple.

**Summary Of The Review:**

I would not recommend the paper for publication at such a top-tier conference as ICLR. The simplicity could be justified, in principle, by the novelty. But this paper is too simple, and the novelty is moderate.

---

> ### Author Response · Authors · 2021-11-13
> **[Part 2 / 2] Minor clarification and summary**
>
> ### Minor
>
> We provide a minor clarification on
>
> > The third method, Structured derivatives, is based on a simple identity (9) that simplifies the computation of NTK over weights of the same layer
>
> We would like to point out that Structured derivatives use several different identities for different kinds of structure in different primitives, and also allow to simplify the computation of the NTK contributions of different layers, if a single weight is re-used in different primitives (see Supplementary Material). While our approach is indeed illustrated by Equation (9), it does not reduce to only applying that specific identity, and is a much more general system of simplifying tensor contractions present in NTK computations based on the structural knowledge about these tensors.
>
> ### Summary
>
> In conclusion, and in accordance with the [ICLR reviewer guidelines](https://iclr.cc/Conferences/2021/ReviewerGuide), we disagree with your use of simplicity as a metric to judge our paper’s merit (if anything, simplicity is an advantage, not a flaw of the paper). If you have used our simplicity of presentation to infer that producing these results is easy (thereby making them insignificant), we have provided ample evidence to the contrary above. Finally, we do not see any justification in your review for claiming that our technical contributions are neither significant nor novel, given that we have clearly laid out both significance and novelty in the related work (Section 2).
>
> We hope our reply helps clarify the potential utility and impact of our work, and kindly ask you to revisit your recommendation following the ICLR reviewer guidelines. Thank you for your time.
>
>
> ### References
>
> [1] [Meta-Learning with Neural Tangent Kernels](https://arxiv.org/abs/2102.03909), Zhou et. al, 2021.
>
> [2] [Neural Architecture Search on ImageNet in Four GPU Hours: A Theoretically Inspired Perspective](https://arxiv.org/abs/2102.11535), Chen et al., 2021.

---

> ### Author Response · Authors · 2021-11-13
> **[Part 1/2] Core objections**
>
> Thank you for your review. We respectfully disagree with your assessment, and hope that you could revisit your score based on the discussion below.
>
> ### Main discussion
>
> Firstly, we probably all agree that simplicity in and of itself is a strength, not a weakness. Simple results are easy to communicate, understand, verify, and use as building blocks in further research projects. Naturally, [ICLR reviewer guidance](https://iclr.cc/Conferences/2022/ReviewerGuide) does not suggest penalizing simplicity. The most impactful ideas in deep learning that have stood the test of time have been simple (e.g. Backpropagation, Glorot Initialization, Residual Networks, Dropout, ReLU nonlinearities, Batch Normalization, Mixup, etc…). This is not a coincidence; it is in large part their simplicity that has led them to proliferate.
>
> For these reasons, we believe that your assessment of our work based on apparent _“simplicity”_ is inappropriate. Research ought to be judged on its potential impact, its value to the community, and its novelty. We argue that our work is both impactful and novel:
>
> 1. Of the 7 papers using the NTK mentioned in related work (Section 2), we have been in touch with the authors of 4 of them. All were using the baseline Jacobian contraction approach, and high computational cost was a common complaint. We are confident that if we were to reach out to everyone (let us know if you’d like us to), all would have been using the Jacobian contraction, and none would be aware of our NTK-vector products or Structured derivatives proposals. Note that NTK has been attracting a lot of research interest since 2018, and in many applications, (e.g. meta-learning [1], Neural Architecture Search [2; NAS] discussed in Section 2) it is clearly the bottleneck of the computation, as the NTK must be recomputed frequently. Absence of our ideas in prior works is evidence of their novelty.
>
> 1. Excluding experiments/tests/examples, our implementation measures 3190 lines of code (LOC). Perhaps by sacrificing documentation, type annotations, and modularity/usability/readability, you could condense it to 2000 LOC. Even provided with the paper and a detailed guide, this would still be difficult to implement, even for seasoned researchers and engineers experienced with JAX (notably, Structured derivatives involve writing a custom Jaxpr interpreter to detect and handle structure in arbitrary JAX primitives and computational graphs). It is completely unrealistic to expect that researchers would be able to do it “on the fly”, and hence wouldn’t benefit from our work. On the contrary, we are certain it will save many researcher- and GPU-years if shared with the ICLR audience, and thereby argue that it is impactful. Note that [ICLR Call For Papers](https://iclr.cc/Conferences/2022/CallForPapers) gives no special treatment for _“ideological / mathematical”_ difficulty, and lists **“implementation issues, parallelization, software platforms, hardware”** as a relevant topic, and you cannot discard it based on your personal preference for mathematics.
>
> 1. Finally, we kindly ask you to keep in mind that good ideas often seem obvious in retrospect, but it doesn’t mean they were easy to come up with. A classic analogy is prime factorization or other NP problems: verifying a prime decomposition is trivial, but coming up with it is NP. Inferring the amount of effort from the simplicity of the result is a tricky and subjective endeavor, which is why we again ask you to judge our work based on novelty and significance (per [ICLR guidelines](https://iclr.cc/Conferences/2021/ReviewerGuide)), and not based on how easy or hard you perceive it has been to produce.
>
>
> Therefore, we take issue with your claim that
>
> > Technical Novelty And Significance: 1: The contributions are neither significant nor novel.
>
> We have presented two novel, absent from prior literature, algorithms that reduce the asymptotic complexity of the NTK computation. We have demonstrated practical, orders of magnitude speed-ups and memory savings. We have packaged, documented, and open-sourced our work for everyone to accelerate their research projects. We have listed numerous examples of such projects and applications that would be impacted by our findings in Section 2. On what grounds do you conclude that our work is neither novel nor significant?

---

> ### Author Response · Authors · 2021-11-20
> **Discussion ends in 3 days**
>
> Dear reviewer,
>
> The discussion period ends in 3 days, so we wanted to kindly remind you to reply to our rebuttal in previous messages.
>
> We additionally wanted to clarify:
> > The paper describes three methods for computing the neural tangent kernel for a given a batch of vectors and a feed-forward neural network.
>
> Our methods apply to any differentiable functions, not only feedforward networks (as stated in our abstract and throughout the text). Our algorithms provide orders of magnitude speed-ups and memory savings on complex ImageNet models like ResNets, Vision Transformers, MLP-mixers, and hybrids, as demonstrated in Figures 2 and 4. Since you didn’t specify in which way exactly our paper is “too simple” or how it could be improved, we wanted to reiterate this, to make sure there’s no misunderstanding.
>
> Thank you again for your time.

---

### Official Review · Reviewer_qdx1 · 2021-11-01

**Correctness:** 3
**Technical Novelty And Significance:** 2
**Empirical Novelty And Significance:** 4
**Recommendation:** 6
**Confidence:** 3

**Main Review:**

Strengths:
- Computing the NTK is a popular area at the moment and the open-source code should make some experiments possible that were not previously for some researchers.
- The approaches proposed exploits nice tricks relating to autodifferentiation (like jvps and vjps) and structure in the computation graphs, which seems tailor made for JAX.
- The experiments are convincing are back up the author's analysis of the advantages of their methods
- The paper is generally very well written and clear, I especially like the colour-coded variables like N,O,P etc

Weaknesses (most of these are not huge or beyond the current scope of the paper):
- The actual technical contribution is not huge, and the paper is more about implementing known ideas from autodiff, linear algebra, and particular aspects of NN structure.
- While improvements on previous implementations is great (and i suspect is now optimal?), it seems these improvements are still far more expensive that standard sgd training (quadratic in batch size, and output size vs linear), which will hamper the impact and usage of this work. For example, 1 second to compute the NTK of a resnet 18 on a v100 for a single pair of imagenet inputs is not ideal. Also is average pooling applied here? I recall average pooling making computational requirements very tricky for CNNs. This may seem a bit unfair a criticism but I feel it is valid given that this is a code paper.
- While it is true to JVP and vjp are on the same order of complexity as forward passes, they are strictly more expensive (i believe a jvp is on the order of 3 forward passes), so while the orders you provide e.g. in Table 1,2 are true, in practice the methods will translate to be slightly slower that simply taking the values in Table 1,2 as gospel. I think it would be good to mention this.
- I think JAX is nice, but wondering if the authors also be implementing their approaches in other frameworks like PyTorch? A lot of non-google/deepmind ppl I know prefer PyTorch to a framework like JAX, and so I believe the impact of the proposed approaches would be greater if so. I realise there are properties with the way that autograd is written in pytorch that makes this more difficult, but I feel like an implementation in just JAX serves to benefit a subset of the research community not the whole, particularly one with a commercial motive.

Minor points:
- The legend in fig 1 left blocks the graph
- When you write mathematically in the NTK in the abstract and eq 1, it's really not clear what f, theta, x are? Nor what dimensions that have. I know it's standard notation but some may not have seen this. You can probably get away with it in the abstract for space, but in eq 1 you should spell these out imo.


**Summary Of The Paper:**

This paper studies the practical compute and memory requirements to computing the Neural Tangent Kernel (NTK), introducing two new approaches to doing so for standard NN primitives (structured derivative and NTK-vector product) which each have advantages (in terms of variables like batch size, output dim) over the naive jacobian contraction method. The authors provide experiments demonstrating the advantages of their approaches across a range of architectures and hardware. The authors provide open-source code which seems to be integrated neatly with the JAX and Neural Tangents frameworks.

**Summary Of The Review:**

Nice code which should be used by some researchers. Using methods/ideas that are established. Concerns that calculating NTK is still too expensive (which it is by nature, not fault of this paper) to prohibit some researchers and also that uptake in using the code will be only by a subset of the community.

---

> ### Author Response · Authors · 2021-11-13
> **[Part 3/3] References**
>
> ### References
>
> [1] [Meta-Learning with Neural Tangent Kernels](https://arxiv.org/abs/2102.03909), Zhou et. al, 2021.
>
> [2] [Neural Architecture Search on ImageNet in Four GPU Hours: A Theoretically Inspired Perspective](https://arxiv.org/abs/2102.11535), Chen et al., 2021.
>
> [3] [Decomposing reverse-mode automatic differentiation](https://arxiv.org/abs/2105.09469), Frostig et al., 2021.
>
> [4] [MetaInit: Initializing learning by learning to initialize](https://papers.nips.cc/paper/2019/hash/876e8108f87eb61877c6263228b67256-Abstract.html), Dauphin et al., 2019.
>
> [5] [Neural Architecture Search on ImageNet in Four GPU Hours: A Theoretically Inspired Perspective](https://arxiv.org/abs/2102.11535), Chen et al., 2021.
>
> [6] [When Vision Transformers Outperform ResNets without Pre-training or Strong Data Augmentations](https://arxiv.org/abs/2106.01548), Chen et al., 2021.
>
> [7] [Wide Neural Networks of Any Depth Evolve as Linear Models Under Gradient Descent](https://arxiv.org/abs/1902.06720), Lee et al., 2019.

---

> > ### Comment · Reviewer_qdx1 · 2021-11-19
> > **Thanks for the response**
> >
> > Thank you to the authors for taking the time to respond to my review, and for the clarifications to my understanding. As I described in my initial rebuttal, my concerns were mostly minor or out of reasonable scope of the work. The authors have largely satisfied my concerns (or at least what I could reasonably expect from a 2 week rebuttal) and so I am maintaining my score.
> >
> > Specific point:
> > 1. *Technical contribution* I appreciate that the ideas proposed may not have been published in the literature, but I still think that some of these ideas (or related ideas) were established. For example, with square loss the hessian becomes the NTK, and so computing NTK-vector products is like computing hessian-vector products. Thus, Section 4.2 of these lecture notes https://www.cs.toronto.edu/~rgrosse/courses/csc2541_2021/readings/L02_Taylor_approximations.pdf describes essentially the NTK-vector product method in eq 10, assuming that the grad operation is doing a vjp which is common in practice as backwards AD is prefered for NNs. Likewise, the idea of structured derivatives for MLPs is essentially the crux of the idea that is leveraged by KFAC (Martens and Grosse 2015) https://arxiv.org/abs/1503.05671 for second order optimisation. The structured derivates for general functions is something that I haven't seen before though, so nice one.

---

> > > ### Author Response · Authors · 2021-11-20
> > > **Thank you for the quick response! Some clarifications**
> > >
> > > Thank you for your quick reply! We’re happy we satisfied most of your concerns.
> > >
> > > > For example, with square loss the hessian becomes the NTK, and so computing NTK-vector products is like computing hessian-vector products. Thus, Section 4.2 of these lecture notes https://www.cs.toronto.edu/~rgrosse/courses/csc2541_2021/readings/L02_Taylor_approximations.pdf describes essentially the NTK-vector product method in eq 10, assuming that the grad operation is doing a vjp which is common in practice as backwards AD is prefered for NNs.
> > >
> > > Thank you for pointing out this very interesting connection, we will include a discussion of the relationship between the NTK and the Hessian in our next revision. While the connection between the NTK and the Hessian is tantalizing, we point out that the Hessian does not become the NTK for the case of MSE loss (see, for example, page 21 in the notes you referenced).
> > >
> > > * For nonzero values of the MSE loss, the Hessian has two terms $H = H_0 + H_1$ where $H_0 = J^TJ$ and $H_1 = e \left(\partial^2f/\partial\theta^2\right)$ where $e$ is the residual error (see [1, Section 2] or Equation (13) in your referenced notes).
> > > * Even in the case that the MSE loss is zero, the Hessian and the NTK are dual to one another in the sense that $H_0 = J^TJ$ but $NTK = JJ^T$). This has important computational implications since we are able to instantiate the NTK explicitly for models whose Hessian would be so large as to be intractable. Therefore, while NTK- and the Hessian-vector products both can be described by composing AD operations, the actual computations are quite different (notably, in the NTK expression, both VJP and JVPs are called on $f$, and `grad(J)` would be the input tangent to the JVP, not the function to differentiate).
> > >
> > >
> > > Outside of the zero-MSE case, we also note that, conceptually, it is more natural to think of the Hessian as a composition of AD operations. In particular, the Hessian is defined as the Jacobian of the gradient/Jacobian, i.e. a _composition_ of two differentiation operations, and is hence natural to implement as the composition described in your note, and as it is implemented in JAX (in fact, we don’t know if there are _any other ways_ to obtain it). In contrast, the NTK is defined as the _matrix product_ of two Jacobians $J  J^T$, and it was naturally implemented in all prior works respectively (i.e. through Jacobian contraction). Therefore we believe implementing it through a composition of JVPs/VJPs was a less straightforward application of AD than in the case of the Hessian.
> > >
> > >
> > > > Likewise, the idea of structured derivatives for MLPs is essentially the crux of the idea that is leveraged by KFAC (Martens and Grosse 2015) https://arxiv.org/abs/1503.05671 for second order optimisation.
> > >
> > > Thank you for pointing out this similarity, we will also mention it in the next revision. While indeed both K-FAC and us apply the mixed-product property, we argue that they are applied in different contexts. K-FAC uses it for the purpose of a cheaper matrix inversion. Due to cubic scaling, it is quite straightforward that a kronecker-factorized matrix is cheaper to invert. We, however, use it for a cheaper contraction, where the mixed-product property on its own is not enough, i.e. Equation (9) does not follow from mixed-product only (mixed product on its own would allow to reduce the cost of Equation (7) from $O W^3 + O^2 W^2$ to only $W^3 + O W^2 + O^2 W$, but we reduce it to $O^2 W$ by symbolically simplifying expressions with the identity matrix. In general, we use a more generic identity described in Section C.2 (and an even more generic tensor contraction when accounting for the batch dimensions $N$), and leverage the fact that one of the inner factors is an identity matrix (Section C.3), as well as the inside-out contraction order that is not possible with standard AD tools. All these components (direct-sum matrix identity from Section C.2, symbolic identity matrix, optimal contraction order) are crucial for Structured derivatives, and mixed-product alone would not suffice.
> > >
> > > In summary, we definitely agree that we use simple linear algebra and AD tools, but we still believe that identifying computations where they could be applied for our purposes is a significant contribution that does not follow straightforwardly from any particular prior application like the Hessian or K-FAC. We do recognize the similarity, and will mention the connections in the next revision, but we still wanted to point out the above differences, since we believe the connections are less immediate than they may appear.
> > >
> > >
> > > > The structured derivates for general functions is something that I haven't seen before though, so nice one.
> > >
> > > We’re glad you liked it! Thank you again for your time and for getting back to us quickly.
> > >
> > >
> > > ### References
> > >
> > > [1] [Geometry of Neural Network Loss Surfaces via Random Matrix Theory](http://proceedings.mlr.press/v70/pennington17a.html), Pennington et al., 2017.

---

> > > > ### Comment · Reviewer_qdx1 · 2021-11-20
> > > > **Thanks**
> > > >
> > > > Thanks for the response and clarifications. As you pointed out I was mistaken in the duality between NTK and Hessian for square loss, but I think we are in agreement that these construtions use similar ideas (and this was my original point in my rebuttal that the technical contribution was "not huge", which I still believe). In any case, I see the main contribution of this work as a software contribution that should provide some use to researchers (though as mentioned I have concerns beyond the scope of this work as to how impactful it may be).
> > > >
> > > > I also just wanted to point out also that for piecewise linear NNs such as those with ReLU or leaky ReLU (which are commonly used), then $H_1$ in your notation is zero regardless of the residual being zero or not.

---

> > > > > ### Author Response · Authors · 2021-11-20
> > > > > **Thank you, minor clarification**
> > > > >
> > > > > Thank you again for your quick reply. As a minor clarification,
> > > > >
> > > > > > I also just wanted to point out also that for piecewise linear NNs such as those with ReLU or leaky ReLU (which are commonly used), then $H_1$ in your notation is zero regardless of the residual being zero or not.
> > > > >
> > > > > Please note that $H_1$ will remain non-zero for these networks, and even for linear networks with 1 or more hidden layers. Such NNs are piecewise linear in their inputs $x$, but not in their parameters $\theta$, and an $n$-hidden layer NN would be an $(n+1)$th order polynomial in their parameters, hence $\partial^2 f /\partial \theta^2$ will be non-zero for $n \geq 1$.
> > > > >
> > > > > Thank you again for your quick replies and active participation, we greatly appreciate it.

---

> > > > > > ### Comment · Reviewer_qdx1 · 2021-11-20
> > > > > > **cheers**
> > > > > >
> > > > > > Yep you are right, cheers for the correction. I was getting confused by the fact that the layerwise block diagonal elements of $H_1$ will be 0 for piecewise linear NNs.

---

> ### Author Response · Authors · 2021-11-13
> **[Part 2/3] JAX vs PyTorch and minor points**
>
> > I think JAX is nice, but wondering if the authors also be implementing their approaches in other frameworks like PyTorch? A lot of non-google/deepmind ppl I know prefer PyTorch to a framework like JAX, and so I believe the impact of the proposed approaches would be greater if so. I realise there are properties with the way that autograd is written in pytorch that makes this more difficult, but I feel like an implementation in just JAX serves to benefit a subset of the research community not the whole, particularly one with a commercial motive.
>
> Thank you for bringing this up, we will elaborate on our decision in the next revision. We really like PyTorch and we agree that it would be nice to support it in the future since, as you suggest, many researchers prefer PyTorch. However, there were several reasons why we chose JAX for our initial implementation:
>
> 1. Structured derivatives rely on manual implementation of structure rules for primitives present in the computation graph of a function. JAX has a small primitive set of about 131 primitives, while PyTorch has more than 400 (see [3, section 2], also https://github.com/pytorch/pytorch/blob/master/tools/autograd/derivatives.yaml vs https://jax.readthedocs.io/en/latest/jax.lax.html). Further, by leveraging `jax.linearize`, we reduce our task to implementing structure rules for only _linear_ primitives, of which JAX has only 54 (This follows from the fact that the NTK of a function is equal to the NTK of the linearized function at the same primal parameters $\theta$; see also [3, section 1], for how JAX uses the same insight to not implement all 131 VJP rules, but only implement 54 transpose rules for reverse-mode AD). To our knowledge PyTorch does not have an equivalent transformation, and hence JAX was a natural choice due to the very concise set of primitives that we need to handle.
>
> 1. NTK-vector products critically rely on forward mode automatic differentiation (JVP), and Structured derivatives also use it (albeit it’s not crucial). PyTorch appears to not have an efficient forward-mode AD (see https://pytorch.org/docs/stable/generated/torch.autograd.functional.jvp.html#torch.autograd.functional.jvp – _“The jvp is currently computed by using the backward of the backward (sometimes called the double backwards trick) as we don’t have support for forward mode AD in PyTorch at the moment.”_). This means that, at least for the moment, NTK-vector products cannot be efficiently implemented in PyTorch.
>
> 1. All implementations (even Jacobian contraction) rely heavily on `jax.vmap` (and we believe, in many cases, it is indispensable). While PyTorch has [released](https://github.com/pytorch/pytorch/issues/42368) a prototype of `vmap` in May 2021, it was not available when we started this project (more than a year ago, with initial prototypes dating back all the way to 2019).
>
> For the reasons above, implementing our algorithms in PyTorch or other frameworks would be a challenging undertaking. For the moment, we would like to focus more on (a) implementing an API for NTK-vector products (supporting all 3 implementations), and (b) looking into a broader set of computations that might benefit from the Structured derivatives ideas. However, we would not rule out a PyTorch implementation in the future if there were demand for it (and if forward-mode AD & linearization made it into the library). In the meantime, something like https://github.com/dmlc/dlpack could be the best short-term option, as it appears to enable zero-cost data transfer between JAX and PyTorch, allowing users to mix the two frameworks. Alternatively, PyTorch has new XLA bindings and it is not out of the question that models could be round tripped from PyTorch to JAX (this is currently done with TensorFlow using the [`jax2tf`](https://github.com/google/jax/tree/main/jax/experimental/jax2tf) transformation).
>
>
> > The legend in fig 1 left blocks the graph
>
> Thank you, we will fix this in the next revision.
>
>
> > When you write mathematically in the NTK in the abstract and eq 1, it's really not clear what f, theta, x are? Nor what dimensions that have. I know it's standard notation but some may not have seen this. You can probably get away with it in the abstract for space, but in eq 1 you should spell these out imo.
>
> Good point, we will add a notation explanation around Eq. (1) in the next revision.

---

> ### Author Response · Authors · 2021-11-13
> **[Part 1/3] On technical contributions and computational costs**
>
> Thank you for your careful and encouraging review, as well as specific constructive comments! Please find our replies below.
>
> > The actual technical contribution is not huge, and the paper is more about implementing known ideas from autodiff, linear algebra, and particular aspects of NN structure.
>
> We respectfully disagree with this point. While our paper uses existing AD tools and linear algebra equations, to our knowledge computing the outer product of Jacobians via NTK-vector products or Structured derivatives is not a _“known idea”_ – it does not seem to exist in any prior work. From private correspondence with some of the authors of papers from Section 2 (related work), we believe that no one was aware of our approaches, and are quite confident that all prior works were using the Jacobian contraction method (if you’d like, we can reach out to all the authors from Section 2 to confirm). Therefore, we believe it is unfair to label our methods as _“known”_ or _“established”_ ideas, even if in retrospect they might seem obvious. Interest in the NTK has been growing rapidly since 2018, and many of the works we mention in Section 2 are certainly bound by its cost (notably, meta-learning [1] and Neural Architecture Search [2; NAS], where the NTK has to be evaluated very frequently), so we believe if these ideas were known or easy, they would’ve been present in the literature by now.
>
>
> > While improvements on previous implementations is great (and i suspect is now optimal?) it seems these improvements are still far more expensive that standard sgd training (quadratic in batch size, and output size vs linear), which will hamper the impact and usage of this work. For example, 1 second to compute the NTK of a resnet 18 on a v100 for a single pair of imagenet inputs is not ideal.
>
> Thank you for the comment, we will add discussion on this subject in the next revision. While this concern is valid, it applies to almost all papers studying infinite width neural networks or other various applications of the finite width NTK – a large and active field. A primary contribution of our work is that it makes the existing body of literature on the NTK much less impractical.
>
> We also point out some applications where our methods can be useful without instantiating the whole $N O \times N O$ NTK:
>
> 1. Our proposed algorithms can be adapted to provide efficient implementations of NTK-vector products without instantiating neither the entire NTK nor the entire Jacobian (as well as vector-NTK products, and vector-NTK-vector products). These can then be used in a variety of tasks, for example evaluating the trainability of the network by computing the NTK’s top eigenvalue [7] via a power method, or performing kernel regression with NTK via conjugate gradients. Providing an API for NTK-vector products based on our algorithms is in our immediate plans!
>
> 1. For the purposes of computing the conditioning of neural networks, it suffices to compute the NTK over small batches of the data. In this case, the computation is orders of magnitude less costly than SGD training. For example in [4, 5, 6] the authors estimate the conditioning by computing an approximation to the NTK on $N$ equal to  128, 32, and 48 examples respectively.
>
> 1. Similarly, [1] use a small batch size of $N = 25$, allowing to actually save time by backpropagating through the NTK kernel regression instead of the SGD training loop inside a MAML-like setting.
>
> 1. Finally, in the above settings authors often use an approximation to the NTK that is comparable to setting $O = 1$ in terms of compute cost. While an important contribution of our work is to allow computing the full NTK using our fast methods, if needed they can also be combined with the $O = 1$ approximation, and still provide a substantial speedup relative to prior work.
>
>
> > Also is average pooling applied here? I recall average pooling making computational requirements very tricky for CNNs. This may seem a bit unfair a criticism but I feel it is valid given that this is a code paper.
>
> Pooling is applied, and all other operations used in ResNets/Vision Transformers/MLP Mixers or hybrids are used in Figures 2 and 4. All models were taken from https://github.com/google/flax/blob/main/examples/imagenet/models.py and https://github.com/google-research/vision_transformer without any modifications. Pooling is indeed very costly for _infinite width_ NTK, but for finite width, it does not pose an issue.
>
>
> > While it is true to JVP and vjp are on the same order of complexity as forward passes, they are strictly more expensive (i believe a jvp is on the order of 3 forward passes), so while the orders you provide e.g. in Table 1,2 are true, in practice the methods will translate to be slightly slower that simply taking the values in Table 1,2 as gospel. I think it would be good to mention this.
>
> Thank you, we will add this in the text and Tables in the next revision.

---

### Official Review · Reviewer_tZAo · 2021-11-02

**Correctness:** 3
**Technical Novelty And Significance:** 1
**Empirical Novelty And Significance:** 3
**Recommendation:** 6
**Confidence:** 4

**Main Review:**

- This paper is well-written and easy to understand. The novelty of this work is fairly weak. All results come from simple computations of automatic differentiation, and the reduced cost of NTK follows from amortizing simple linear operations, which is not surprising at all. However, these weaknesses might be covered by their open-source implementation, as it can be significantly important and useful to future works.

- The infinite-width NTK can be exactly computed without Jacobians (Arora et al., 2019) and it can be much efficient than Jacobian-based approaches. Does the finite-width NTK have concrete benefits compared to the infinite-width NTK?

- Minor Issue:
  - It would be great if more details of Automatic Differentiation (AD) operations to derive memory costs of JVP/VJP are provided.
  - As assumed $O=\mathcal{O}(LW)$, the time and memory costs in Section 3.2~3.4 can be reduced without $O$.
  - The method names in Figure 1 are too small to identify. It would be great to make larger  font size.



**Summary Of The Paper:**

This paper studies an in-depth analysis of runtime and memory requirements for computing the finite-width NTK. The authors analyze computing costs of Jacobian-vector products (and vice versa) for both fully-connected and convolutional neural networks. They also improve the NTK computation cost by leveraging the structure of neural networks resulting in drastic speedup and memory saving. Finally, they make all their implementations open-source based on the JAX library.

**Summary Of The Review:**

The review score is all about the battle of lack of novelty versus the impact of implementations. I judge this work to be under the bar of acceptance for now but am willing to raise it depending on the author's feedback on the importance of computing finite width NTK.

---

> ### Author Response · Authors · 2021-11-13
> **[Part 5/5] References**
>
> ### References
>
> [1] [On Exact Computation with an Infinitely Wide Neural Net](https://arxiv.org/abs/1904.11955), Arora et al., 2019.
>
> [2] [Wide Neural Networks of Any Depth Evolve as Linear Models Under Gradient Descent](https://arxiv.org/abs/1902.06720), Lee et al., 2019.
>
> [3] [Meta-Learning with Neural Tangent Kernels](https://arxiv.org/abs/2102.03909), Zhou et. al, 2021.
>
> [4] [Neural Architecture Search on ImageNet in Four GPU Hours: A Theoretically Inspired Perspective](https://arxiv.org/abs/2102.11535), Chen et al., 2021.
>
> [5] [When Vision Transformers Outperform ResNets without Pre-training or Strong Data Augmentations](https://arxiv.org/abs/2106.01548), Chen et al., 2021.
>
> [6] [Neural Tangents: Fast and Easy Infinite Neural Networks in Python](https://arxiv.org/abs/1912.02803), Novak et al., 2019.
>
> [7] [Flax: A neural network library and ecosystem for JAX](https://github.com/google/flax), Heek et al., 2020.
>
> [8] [Haiku: Sonnet for JAX](https://github.com/deepmind/dm-haiku), Hennigan et al., 2020.
>
> [9] [Infinite attention: NNGP and NTK for deep attention networks](https://arxiv.org/abs/2006.10540), Hron et al., 2020.
>
> [10] [A Mean Field Theory of Batch Normalization](https://arxiv.org/abs/1902.08129), Yang et al., 2019.
>
> [11] [Tensor Programs II: Neural Tangent Kernel for Any Architecture](https://arxiv.org/abs/2006.14548), Yang, 2020.
>
> [12] [Graph Neural Tangent Kernel: Fusing Graph Neural Networks with Graph Kernels](https://arxiv.org/abs/1905.13192), Du et al., 2019.
>
> [13] [Evaluating Derivatives](https://epubs.siam.org/doi/book/10.1137/1.9780898717761?mobileUi=0), Griewank & Walther, 2008.
>
> [14] [MetaInit: Initializing learning by learning to initialize](https://papers.nips.cc/paper/2019/hash/876e8108f87eb61877c6263228b67256-Abstract.html), Dauphin et al., 2019.
>
> [15] [An iteration method for the solution of the eigenvalue problem of linear differential and integral operators](https://www.cs.umd.edu/~oleary/lanczos1950.pdf), Lanczos, 1950.
>
> [16] [The asymptotic spectrum of the Hessian of DNN throughout training](https://arxiv.org/abs/1910.02875), Jacot et al., 2019.
>
> [17] [Tilting the playing field: Dynamical loss functions for machine learning](http://proceedings.mlr.press/v139/ruiz-garcia21a.html), Ruiz-Garcia et al. 2021.
>
> [18] [Geometry of Neural Network Loss Surfaces via Random Matrix Theory](http://proceedings.mlr.press/v70/pennington17a.html), Pennington et al., 2017.
>
> [19] [The large learning rate phase of deep learning: the catapult mechanism](https://arxiv.org/abs/2003.02218), Lewkowycz et al., 2020.
>
> [20] [Finite Versus Infinite Neural Networks: an Empirical Study](https://arxiv.org/abs/2007.15801), Lee et al., 2020.

---

> > ### Comment · Reviewer_tZAo · 2021-11-22
> > **After the author response**
> >
> > I would like to thank the authors for their detailed feedback on my comments. Their feedback is very satisfactory as they have answered the various importances of F-NTK beyond the I-NTK and updated all minor issues. The updated paper looks even better. Regarding the novelty, I agree with the authors that the implementations can be of significant standard in the ICLR review process. And this work includes the great effort of F-NTK implementation and I believe this will pave the way for many future works required to the NTK in practice. Hence, I raise my score to the weak acceptance. Appreciate again the authors' great effort in writing the feedback.

---

> > > ### Author Response · Authors · 2021-11-22
> > > **Thank you**
> > >
> > > Thank you for recognizing our contributions, and for your specific and constructive feedback that allowed us to improve the paper!

---

> ### Author Response · Authors · 2021-11-13
> **[Part 4/5] Novelty & conclusion**
>
> ## Novelty
>
> Finally, we would like to argue that your assessment of our technical contributions might be too harsh, and kindly ask you to reconsider this score as well.
>
> > Technical Novelty And Significance: 1: The contributions are neither significant nor novel.
>
> We have presented two novel, absent from prior literature, algorithms that reduce the asymptotic complexity of the NTK computation. We have demonstrated practical, orders of magnitude speed-ups and memory savings. We have packaged, documented, and open-sourced our work for everyone to accelerate their research projects. We have listed numerous examples of such projects and applications that would be impacted by our findings in Section 2 (and some more in our response above). Therefore, we argue that our work is both novel and significant.
>
> > The novelty of this work is fairly weak. All results come from simple computations of automatic differentiation, and the reduced cost of NTK follows from amortizing simple linear operations, which is not surprising at all.
>
> Firstly, albeit anecdotally, we disagree with your characterization of our results as _“not surprising at all”_. It was not at all obvious to us or anyone we discussed our ideas with, that the NTK computation could be sped up beyond the Jacobian contraction. One colleague of ours was exceedingly surprised by how fast Structured derivatives were. As we mention in Section 3.4 and show in our plots, in certain settings Structured derivatives allow to compute the Jacobian contraction faster and with less memory than it takes to compute the Jacobians themselves – isn’t this result at least somewhat surprising?
>
> In addition, to our knowledge no prior work has come up with such ideas in the 3 years since the introduction of NTK, which is another bit of evidence towards our contributions being not that obvious.
>
> We would also like to point out that [ICLR reviewer guidelines](https://iclr.cc/Conferences/2021/ReviewerGuide), as well as the inline hints$^1$ next to the _"Technical Novelty and Significance"_ score do not suggest judging the work based on how simple it seems or the degree to which the result is surprising. In fact, the impact of work is often proportional to its apparent simplicity in retrospect.
>
> $^1$ [Specifically, the hint says _“For this question, contributions are technical in nature, including new models, techniques, or theoretical insights”._ We argue that we provide a theoretical insight into the computational complexity of the NTK.]
>
> In addition to the absence of our ideas in prior works, another piece of evidence in favor of novelty is that our codebase, excluding experiments/tests/examples, contains more than 3000 lines of code. It was not easy to produce, nor was it obvious that it was even possible to implement in the first place (notably, Structured derivatives for arbitrary primitives / computational graphs). Since [ICLR Call For Papers](https://iclr.cc/Conferences/2022/CallForPapers) lists **“implementation issues, parallelization, software platforms, hardware”** as a relevant topic, we ask you to not overlook the technical novelty of our engineering effort in your assessment as well. We believe sharing our implementation with the ICLR audience will save many researcher- and GPU-years.
>
> ## Conclusion
>
> > I judge this work to be under the bar of acceptance for now but am willing to raise it depending on the author's feedback on the importance of computing finite width NTK.
>
> We hope that the above discussion has convinced you of the finite width NTK significance, as well as of the novelty of our approach, and kindly ask you to revisit your score accordingly. If not, please let us know of your further questions/concerns, and we will be happy to continue the discussion. Thank you for your time and consideration!

---

> ### Author Response · Authors · 2021-11-13
> **[Part 3/5] Infinite width NTK (I-NTK) limitations**
>
> In contrast, the I-NTK is the limit (in the infinite width) of the expectation of the F-NTK over i.i.d. Gaussian $\theta \sim \mathcal{N}\left(0, I_{P}\right)$. This immediately imposes numerous limitations:
> 1. As discussed above, the I-NTK does not depend on any specific realization of parameters $\theta$, and thus cannot be used to directly optimize or interpret network parameters [3, 14].
>
> 1. The I-NTK requires the width of the network to be very large to give relevant predictions, which makes it less suitable for NAS [4] or model selection [5], where finite width effects are often important.
>
> 1. In order to exist, the I-NTK requires the network to have a concept of width to begin with and various conditions on the initial distribution of $\theta$, the nonlinearities, and the topology need to be satisfied. For a _closed-form_ solution to the I-NTK to exist, the network further has to be built out of a very limited, hand-selected, number of primitives that admit closed-form computations of certain Gaussian integrals. We conjecture that the space of functions admitting a closed-form infinite width NTK is a measure-zero subset of the space of all differentiable functions (admitting the F-NTK), and also a very small subset of functions used in practice. In the long term, the I-NTK is unlikely to scale to the enormous variety of new architectures introduced by the research community each year (due to stringent conditions for convergence + large width + unknown conditions necessary for the Gaussian integrals to have a closed-form solution), while the F-NTK is defined and computable with our methods for any differentiable function.
>
> 1. As an example, we highlight that the work you have referenced for the I-NTK [1], was [implemented](https://github.com/ruosongwang/cntk) in CUDA, for a very specific all-conv, ReLU-only architecture. Adapting it to even slightly different conditions would be either very laborious due to necessary code changes (e.g. strided convolutions) or impossible due to lack of closed-form limit (e.g. using max pooling instead of average pooling, or Tanh instead of ReLU, etc). As an improvement, [6] allows to compute the I-NTK for a range of  architectures, but _only as long as it is built using their framework, out of the limited set of building blocks provided by the authors_. You cannot take an arbitrary model built in a different library (e.g. Flax [7], Haiku [8], etc) and compute its I-NTK. Instead, you would have to rewrite the model from scratch using [6] (while making sure it’s built out of the right building blocks admitting the closed-form limit, which will generally not be the case). In contrast, our methods require absolutely zero change in response to any modifications to the network architecture, and work with any provided function built in any neural network library, akin to generic JAX function transformations like `jax.jit` or `jax.vmap`.
>
> 1. Components that do not admit an efficient closed form I-NTK include
> 	1. Attention with standard parameterization [9].
> 	1. Max-pooling.
> 	1. Sigmoid, Tanh, and many other nonlinearities.
> 	1. Normalization layers that contract over finite dimensions (e.g. BatchNorm) can be computed [10], but only for very specific cases (linear layers or ReLU layers) and the computational cost is (at least) cubic in the size of the contracted dimension.
> 	1. Networks with nontrivial weight sharing can often not be computed in closed form. When they can be computed, the computational cost is usually exponential in the depth of the network or sequence length [11].
>
> 1. Finally, as we have mentioned above, for most generic architectures I-NTK is very time and memory costly to compute (quadratic scaling with the number of pixels/tokens $D$, see for instance [1, 9, 12]), while finite width is more tractable (linear in $D$ time and memory).
>
> In conclusion of this part, F-NTK is a very generic concept that describes the behavior of the linearization of any differentiable function $f$ around any parameters $\theta$, and our methods allow to efficiently compute it. In contrast, I-NTK is a much more specialized object, in terms of its scope/applicability (need to have width; need width to be very large; need parameters to be i.i.d. Gaussian; etc.), tractability (need to have various conditions satisfied to exist and be computable in closed form; need to use a specific library or write your own implementation manually), and compute cost (likely scales quadratically with the number of pixels/tokens $D$).
>
> Thanks to your comment, we will add discussion of the differences between the F- and I-NTK in the next revision.

---

> ### Author Response · Authors · 2021-11-13
> **[Part 2/5] Finite width (F-NTK) applications**
>
> > Does the finite-width NTK have concrete benefits compared to the infinite width NTK?
>
> Thanks for the great question! Neural networks trained in practice are finite width objects whose training is locally described by the F-NTK. By contrast, the I-NTK is an approximation that is exact only in the limit of infinite width at initialization. As such, the I-NTK has no notion of parameters and cannot be computed during training. Therefore, there are numerous benefits and use cases for the F-NTK.
>
> ### Specific applications
>
> Every reference in Section 2 (related work) refers to the F-NTK, and using an I-NTK in its place wouldn’t make sense. For example, meta-learning and transfer learning [3] requires backpropagating the loss signal into the parameters of the network (which are absent in the I-NTK); in Neural Architecture Search [4; NAS] or model selection [5] the F-NTK predicts trainability/generalization of the particular (finite width) proposal, while the I-NTK would be a worse (infinite width) approximation; in model tuning [14] the F-NTK describes conditioning of the specific finite width network rather than the conditioning in the infinite width limit. We believe the presented use cases in Section 2 and Introduction (meta-learning and transfer learning [3]; NAS [4]; model selection [5]; model tuning [14, 17]; empirical confirmation of the infinite width NTK theory [2, 6]; insight into differences between F- and I-NTK [1]) are significant.
>
> While Section 2 and Introduction gave a number of specific use-cases, we would like to draw attention to a few general principles that we feel motivate the F-NTK. As we mention above, the F-NTK describes the local training dynamics in function space. As such, the F-NTK can be used as a proxy for the Hessian [16, 18] and therefore its conditioning is strongly related to the trainability and predicts the maximum learning rate that a given neural network admits [2, 19]. Unlike the Hessian, however, the F-NTK can be computed without using approximations such as the Lancoz algorithm [15].
>
> One more application that we touched on in the text, but will emphasize in the next revision, is that our methods naturally give rise to very efficient algorithms for computing F-NTK-vector products. This can be used to exactly compute the conditioning of the F-NTK (via the power method), or to perform kernel regression with F-NTK (via conjugate gradients) without explicitly instantiating the entire F-NTK. In contrast, to our knowledge it is not possible to perform efficient I-NTK-vector products without instantiating the whole I-NTK. Implementing an API for efficient NTK-vector products based on our algorithms is in our short term plans.
>
> Here we have argued for the F-NTK, but we do agree with you that the I-NTK is an extremely interesting quantity to study! However, when computing the I-NTK the standard method to check for correctness is to compare the kernel with Monte-Carlo samples of the F-NTK. Thus, efficient computation of the F-NTK is important for unit-testing I-NTK calculations.
>
>
> ### Overall scope
>
> Beyond presenting the specific applications above, we now describe why we believe that the F-NTK is an object of a much wider scope in terms of applicability, as well as mathematical and computational tractability, than the I-NTK. As a consequence, we conjecture that over the coming years, the F-NTK will have a larger impact on neural network research than the I-NTK.
>
> As we discussed above, the F-NTK describes the behavior of the linearization of an arbitrary function, $f(\theta, x)$, around any parameters $\theta$. As long as the function is differentiable, the F-NTK can be computed with our methods and analyzed to infer trainability/generalization/other aspects of $f(\theta, x)$. We stress that the function $f$, inputs $x$, and parameters $\theta$ can be almost anything, and notably parameters $\theta$ can be parameters after a certain amount of training, or parameters of a pretrained model, not necessarily at initialization.
>
> [A small tangent: in fact, as we discussed in Section 4, our algorithms are not specific to neural networks in any way, and are suitable for computing any Tangent Kernel; unlike I-NTK, it does not have to be “Neural”, and we might as well call it F-TK. However, for simplicity we will continue calling it F-NTK throughout.]

---

> ### Author Response · Authors · 2021-11-13
> **[Part 1/5] Finite vs Infinite width NTK compute cost and minor comments**
>
> Thank you for your careful review and specific feedback.
>
> We first briefly address your minor comments, then discuss the importance of computing the finite width NTK (which we abbreviate as **F-NTK**, as opposed to infinite width, **I-NTK**), and finally touch on the question of novelty.
>
> ## Minor
>
> > It would be great if more details of Automatic Differentiation (AD) operations to derive memory costs of JVP/VJP are provided.
>
> Great suggestion, we will add discussion and references on this in the next revision. One brief intuition for why their time costs are comparable to the forward pass (**FP**), is that both can be computed by traversing a computational graph of the same/similar topology to FP (same for JVP; transpose graph for VJP), and the nodes of these JVP and VJP graphs have comparable costs to the respective nodes in the FP graph. Further, JVP does not require extra memory (asymptotically) beyond FP since it traverses the graph in the same order as FP. VJP traverses the graph in the transpose order, but each node of the transpose graph requires primals to perform the node computation, hence memory-wise VJP requires storing pre-activations. A rigorous treatment of the subject can be found in [13, Section 3].
>
>
> > As assumed O=O(LW), the time and memory costs in Section 3.2~3.4 can be reduced without O.
>
> We don’t think this would be appropriate, since it would produce unnecessarily loose bounds. In particular, we would like to specialize our computational complexity to the case where $O = \mathcal O(1)$ since this is a relatively common case (in this case $O$ would still be $\mathcal{O}(L W)$). However, if we had already simplified the complexities by replacing $O$ with $L W$, then, for example, the cost of Jacobian contraction would be bounded by $\mathcal{O}\left(N^2 L^3 W^4\right)$, while a much better bound would be $\mathcal{O}\left(N^2 L W^2\right)$, which we get from our current expression in the text by substituting $O = 1$. Please let us know if we misunderstood you.
>
>
> > The method names in Figure 1 are too small to identify. It would be great to make larger font size.
>
> Thank you, we will fix this in the next revision.
>
>
> ## Significance of finite width NTK (F-NTK)
>
> > The infinite-width NTK can be exactly computed without Jacobians (Arora et al., 2019) and it can be much efficient than Jacobian-based approaches.
>
> Thank you for bringing this up, we will add a section to the paper discussing benefits of the finite width NTK (**F-NTK**) vs the infinite width NTK (**I-NTK**) in practice.
>
> While it is true that the I-NTK _can_ be computed more efficiently in _some_ specific cases, in almost every case of practical interest, the F-NTK is orders of magnitude faster to compute than I-NTK. For example, in convolutional networks with pooling, which is what the paper you reference [1] considered (though note that [2] is the paper which introduces closed form expressions for the I-NTK), the time and memory costs of the I-NTK is quadratic in the total number of pixels $D$ (e.g. $D^2 = 224^4 = 2,517,630,976$ for ImageNet), while our methods are linear in $D$.
>
> This should be true for any generic NN architecture that somehow leverages the spatial structure of the inputs (e.g. Graph Neural Networks, Transformers, MLP Mixers, etc) on most datasets. For a few specific cases (namely, fully-connected networks, CNNs with no pooling, or locally-connected networks) you are correct that I-NTK would be faster than any of our finite width implementations, since it would admit a particular performance optimization and scale at most linearly with $D$ while not incurring any cost due to width $W$ or total number of parameters $P$. But these models are not commonly used in practice, and the respective I-NTKs substantially underperform compared to their more costly counterparts that use pooling [1, 20].
>
> As a specific example, even for $N = O = 1$, it would be impossible to compute the I-NTK for any of the ImageNet models we consider in Figures 2 and 4 on a V100 GPU, since it would require at least 2 * (224^4 * 32) bits = 20 Gb of RAM (V100 has only 16 Gb; factor of 2 is to store the input and output of the I-NTK layer-to-layer kernel propagation).
>
> In terms of time, [1] and [6] report from 0.002 to 0.003 seconds for a single pair of CIFAR10 (32x32) inputs to the I-NTK of a 20-layer ReLU CNN with pooling, on a V100 GPU. In Figure 2 upper-left, we report comparable times for a single pair of _ImageNet_ (224x224 = 50x larger than 32x32) inputs to an F-NTK of a _200-layer ResNet_.
>
> Finally, if considering an identical setup (20-layer 128-wide CNN on binary CIFAR10, as in [1, section B]), we can compute the F-NTK at least as fast as 0.000014 seconds per entry, i.e. at least 100x faster than the equivalent I-NTK.

---

> ### Author Response · Authors · 2021-11-20
> **Discussion ends in 3 days**
>
> Dear reviewer,
>
> Since you indicated that you were
>
> > willing to raise it depending on the author's feedback on the importance of computing finite width NTK.
>
> we wanted to inquire whether you were satisfied with our responses below. If not, please let us know soon so that we could provide more details or address your other questions/concerns before the discussion period ends in 3 days.
>
> Thank you again for your time!

---

### Author Response · Authors · 2021-11-16
**Paper updated**

Dear reviewers, we have uploaded a paper revision addressing you comments and suggestions, and making other improvements:

1. Added notation legend and shapes to Equation 1 in the Introduction.
1. Expanded section 3.1 with explanation and references for JVP/VJP time and memory costs. Added Figure 0 for a visual demonstration.
1. Expanded section 3.1 and Table captions to emphasize that shown are asymptotic time/memory estimates, and in practice all methods also include a multiplicative constant.
1. Increased legend font sizes in all Figures.
1. Put legends outside of Figures 1 and 3 to avoid obstructing the plots.
1. Increased opacity and used different styles for dashed $O = W$ and $W^2$ lines in Figures 1 and 3 to make them visually more distinct.
1. Expanded Figure 5 with a CNN notation visualization.
1. Fixed various typos/mistakes/formatting throughout the text; minor rewording/compression to fit the extended text into 10 pages.
1. Added section I on NTK-vector products and other low-compute/memory settings where our methods can provide speed/memory savings.
1. Added section J on comparing applications and tractability of finite and infinite width NTKs.
1. Added section K on why we chose JAX over other libraries for our initial implementation.
1. Added Section L with the complexity analysis without the assumption of $O = \mathcal{O}(L W)$.

Thank you again for your constructive feedback. Please let us know if you would like to propose other changes.

---

> ### Author Response · Authors · 2021-11-23
> **Second update**
>
> Dear reviewers, we’ve made a few more changes:
> 1. Moved the added discussion on the JVP/VJP costs and Figure 0 (now Figure 6) to the Appendix (now section N), since we’ve found out that [the paper should remain within the 9 page limit this year, with no extra space added during rebuttal](https://iclr.cc/Conferences/2022/CallForPapers).
> 1. Added Section M on the relationship between our methods and computing the Hessian.
> 1. Expanded Section G to discuss the relationship to K-FAC.
> 1. Made more precise notation in Section 4 and Table 2. Added a missing $ N Y $ memory term to all methods.
> 1. Other minor improvements and fixes, and rewording in order to squeeze the paper back into 9 pages.
>
> [Note: we may reorder the Appendix sections into a more logical order after the rebuttal; for now, we’ve been adding them after initial sections in order to not break the section/figure/table numbering]
>
> Thank you again everyone for the detailed feedback and suggestions allowing us to improve the paper!

---

### Decision · Program_Chairs · 2022-01-20

**Decision:**

Reject

**Comment:**

This is an interesting and carefully-presented work which discusses how to implement finite-width NTKs more efficiently.  Overall, the reviews were slightly tending positive, though with a variety of concerns, including some concern that the contribution is not sufficiently substantial.  In my own perusal of the paper, personally I feel it could be made more compelling if (a) more speedups could be considered, including ones with various tradeoffs, for instance via randomized linear algebra, (b) explicit consequences on various prediction tasks, rather than plotting wall-clock times (e.g., as this paper cites many works which tried to use finite-width NTK, and as this paper claims massive speedups, then it will be able to repeat some of those experiments at much larger sizes, which should lead to interesting and valuable larger-scale experiments which ideally have some new phenomena, but are even interesting if they simply confirm the smaller-scale phenomena).  As a separate concern, I second the comments of one reviewer, that part of this paper's contribution is to a single software package, which is moreover listed in the abstract (and not just part of the standard code release, e.g., as a footnote); this feels a little strange, like an announcement of a code release, and further limits the impact to general machine learning researchers (for instance, I feel completing some of my preceding suggestions could result in, say, researchers who use other software feeling eager to re-implement this).  Overall, I urge the authors to continue with their interesting work and aim to resolve these concerns and those of the reviewers.